# Twin-2K-500: A dataset for building digital twins of over 2,000 people based on their answers to over 500 questions

## Abstract

LLM-based digital twin simulation, where large language models are used to emulate individual human behavior, holds great promise for research in business, AI, social science, and digital experimentation. However, progress in this area has been hindered by the scarcity of real, individual-level datasets that are both large and publicly available. To address this gap, we introduce a large-scale, public dataset designed to capture a rich and holistic view of individual human behavior. We survey a representative sample of $N = 2,058$ participants (average 2.42 hours per person) in the US across four waves with over 500 questions in total, covering a comprehensive battery of demographic, psychological, economic, personality, and cognitive measures, as well as replications of behavioral economics experiments and a pricing survey. The final wave repeats tasks from earlier waves to establish a test-retest accuracy baseline. Initial analyses suggest the data are of high quality and show promise for constructing digital twins that predict human behavior well at the individual and aggregate levels. Beyond LLM applications, due to its unique breadth and scale the dataset also enables broad social science and business research, including studies of cross-construct correlations and heterogeneous treatment effects.

## 1 Introduction

The rise of large language models (LLMs) like GPT has sparked interest across disciplines (including marketing, computer science, economics, psychology and political science) in leveraging these tools to create "silicon samples" which may replicate how these humans would behave in response to any stimuli (Arora et al. 2024, Argyle et al. 2023, Brand et al. 2023, Dillion et al. 2023, Goli and Singh 2023, Horton 2023, Li et al. 2025, Park et al. 2023, Qin et al. 2024). If these LLM-simulations can be a faithful substitute for eliciting responses from their human counterparts, the implications for both academics and practitioners are substantial. Academics could use silicon samples for pilot experiments to pinpoint stimuli with significant impact, thus improving the efficiency of theory development and experimental design. Firms could leverage these realistic simulations to explore different ideas and strategies, thereby improving customer insight and product development. Accordingly, in the recent past we have witnessed a large influx of firms offering services leveraging silicon samples for customer insights (e.g., Synthetic Users, Outset AI, Nexxt, Voxpopme, Evidenza, Expected Parrot, Meaningful, xPolls, Ipsos, CivicSync).

While silicon samples may be generated using only demographic information or hypothetical "life stories," a promising approach consists in creating silicon samples that are "digital twins" of real people. Notably, Park et al. (2024) use LLMs to create digital twins of over 1,000 individuals based on transcripts from qualitative interviews, and find that the simulated agents replicated the human participants' responses on the General Social Survey 85% as accurately as participants replicate their own answers two weeks later.

Despite the promise and excitement surrounding digital twins, some uncertainty remains. For example, Brucks and Toubia (2025) show that the answers provided by LLMs may be overly influenced by the architecture of the prompt, such as the labeling or ordering of options in multiple choice questions. Gui and Toubia (2023) show that leveraging LLMs to

simulate experiments may introduce unwanted confounding, due to the difficulty of clearly instructing the LLM how to draw variables not specified in the prompt. Other research (Santurkar et al. 2023, Motoki et al. 2024, Li et al. 2025) suggests LLMs tend to express opinions that are not representative of the (human) population.

Given this background, it is crucial for the academic and practitioner community to validate digital twins in a transparent, reliable, and replicable manner. However, existing datasets present significant limitations that hinder their effectiveness for this purpose. Some datasets are publicly available (e.g., Alattar et al. 2018, Center 2023, Santurkar et al. 2023), but they are not well suited for testing the validity of digital twins because they do not contain behavioral data (e.g., experiments) nor a test-retest accuracy benchmark. Other datasets have this feature, but they are not publicly available (e.g., Park et al. 2024).

In sum, to the best of our knowledge there is no publicly available dataset that combines rich psychological profiles, behavioral data, and demographics from a large, representative sample for the development and testing of digital twin simulations. As a result, researchers often rely on synthetic or proprietary data, which undermines transparency, reliability, and replicability.

To address this gap, we assemble and publicly share an extensive dataset from a representative sample of $N = 2,058$ people who each answered over 500 questions covering a wide range of demographic questions, psychological scales, cognitive performance questions, economic preferences questions, as well as replications of a wide range of within- and between-subject experiments on heuristics and biases taken from the behavioral economics literature. The data was collected across 4 waves of studies lasting on average 2.42 hour per participant in total. Table 3 gives an overview of the measures collected in each wave and Figure 1 illustrates our overall approach.

Figure 1: Overview

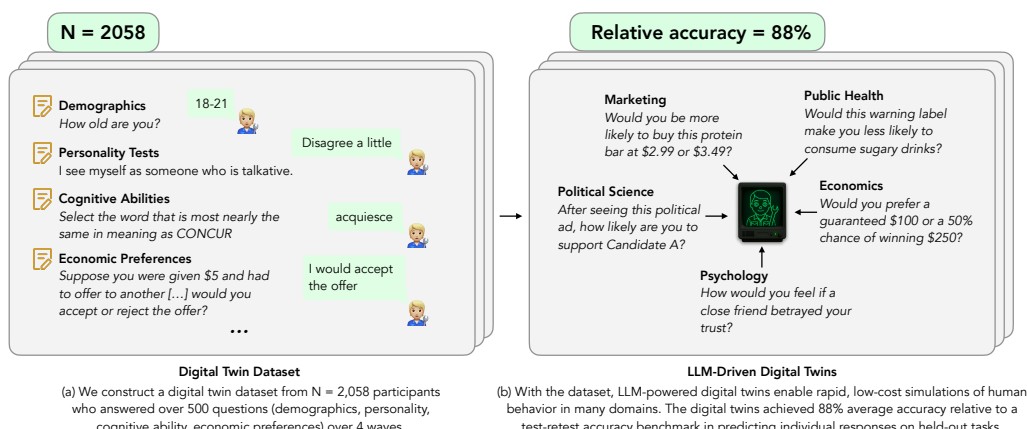

(a) We construct a digital twin dataset from N = 2,058 participants who answered over 500 questions (demographics, personality, cognitive ability, economic preferences) over 4 waves.

(b) With the dataset, LLM-powered digital twins enable rapid, low-cost simulations of human behavior in many domains. The digital twins achieved 88% average accuracy relative to a test-retest accuracy benchmark in predicting individual responses on held-out tasks.

We use the responses to the heuristics and biases questions from waves 1-3 as holdout data, and train the digital twins based on the rest of the data from waves 1-3. Wave 4 repeated the heuristics and biases experiments, providing us with a measure of test-retest accuracy. Future uses of the data may keep the same split, or combine all the data from the 4 waves to create digital twins.

We report encouraging results regarding the quality of the data: correlations between measures have good face validity, we replicate almost all known results from the behavioral economics literature, and the test-retest accuracy is robust.

We also report initial tests of the predictive validity of digital twins constructed using the data. At the individual level, we compute the accuracy of the digital twin predictions on holdout questions, against the test-retest accuracy benchmark as well as a random benchmark. At the aggregate level, we test whether the digital twin simulations replicate

the average treatment effects observed in human data. Throughout this process, we explore the type of behavior that can be predicted with higher vs. lower accuracy by the digital twins, to develop insight into the range of potential applications.

The dataset and code are publicly available.[1] Given the unique scale and breadth of the data, there is also value in the raw results, irrespective of the application to digital twins. We report descriptive statistics and correlations between the dozens of measures we collected. We encourage others to explore heterogeneous treatment effects (Dean and Ortoleva 2019, Stanovich and West 2008).

## 2 Methods

We assembled a wide-range of measures proposed in the business and social science literatures over the past several decades. In addition to 14 demographic questions, we included 19 personality tests that measured 26 constructs over 279 questions, 11 cognitive ability tests (85 questions, 11 measures), 10 economic preferences tests (34 questions, 10 measures). We also replicated 11 between-subject experiments (16 questions) and 5 within-subject experiments (32 questions) from the behavioral economics literature.[2] Finally, we administered the pricing study from Gui and Toubia (2023), which asks participants to make purchase decisions about 40 different products at randomly selected prices. In total, participants answered 500 questions across the first three waves. Wave 4 repeated all within- and between-subject heuristics and biases experiments from the first three waves as well as the pricing study from wave 3 (16+32+40=88 questions in total). Participants were assigned to the exact same condition in wave 4 as they were in waves 1-3 for each of these experiments, providing us with a clean measure of test-retest accuracy. We programmed the studies on Qualtrics, doing our best to replicate the stimuli and measures from the original papers.[3]

We launched wave 1 on Prolific on 01/29/2025, targeting 2,500 representative US respondents (sampled by age, sex, and ethnicity). Participants received $7 for completing wave 1.[4] They were informed that this was the first of four waves and they would earn a $10 bonus for completing all waves (with comprehension checks to ensure understanding). We received 2,509 complete responses. The following week (02/04/2025), we invited these 2,509 participants to wave 2 ($7), receiving 2,263 complete responses. The next week (02/11/2025), we invited them again for wave 3 ($7), receiving 2,252 complete responses. waves 2 and 3 were closed the next week. On 02/25/2025, we invited the 2,154 participants who had completed waves 1–3 to wave 4 ($6; two-week delay since wave 4 repeated previous measures), receiving 2,058 complete responses and closing the wave after one week. Those completing all four waves received an additional $10 bonus, totaling $37.

These 2,058 participants who completed all four waves constitute our final sample. Among our final sample, the average response time was 43.88 minutes for wave 1 (std=19.26), 45.31 minutes for wave 2 (std=19.24), 32.66 minutes for wave 3 (std=15.68), and 24.09 minutes for wave 4 (std=12.51). The average total time across all 4 waves was 145.47 minutes (std=56.10).

---

[1]The dataset is publicly available and the link is hidden for anonymization purpose, and the LLM simulation code can be found at https://anonymous.4open.science/r/Digital-Twin-Simulation-04DF/. (A non-anonymized github repository will be linked upon acceptance).

[2]We included all experiments in Stanovich and West (2008) who study "some of the most classic and well studied biases in the heuristics and biases literature," as well as false consensus which allowed us to both capture participant's opinions on a range of issues and to test another well-known bias.

[3]We made minor adjustments to reflect cultural and societal changes (e.g., in the mental accounting scenarios from Thaler (1985) we replaced "Mr. A" and "Mr. B" with "Person A" and "Person B," and in the sunk cost experiment of Stanovich and West (2008) we replaced video rental stores with coffee shops).

[4]We pre-tested each wave to estimate response time and adjusted compensation accordingly.

## 3 Data

Section 5 presents initial results of the performance of digital twins created with the data. Those results are subject to change as researchers explore optimal ways to create digital twins, and the current results may be viewed as providing a lower bound on performance. In the current section, we instead focus on exploring the *intrinsic* quality of the data.

Table A1 reports demographic characteristics of our sample. While our digital twins are created based on the raw responses, it is also informative and potentially useful to extract the measures corresponding to these questions (e.g., the extraversion score is measured by averaging 8 questions from the Big Five battery of questions). Across waves 1-3, we collected 47 measures capturing personality traits, cognitive abilities and economic preferences. Studying the correlations between these measures is of general interest to business and social science scholars and practitioners, above and beyond the question of digital twins. Web appendix A3 details the construction of these measures from the raw data, and Table A2 reports summary statistics for the individual-level measures collected in the study. We compute a total of 1,326 pairwise correlations between the 47 measures listed in Table 3 and 5 demographic characteristics. We apply the Bonferroni correction and consider a correlation as significant if the $p$-value is below $\frac{0.05}{1326}$. This gives us 509 pairs of measures with significant correlations. We cannot report them all here, and instead report in Table A3 10 examples of correlations that are particularly high and/or noteworthy. These correlations all have good face validity, which suggests the data are of good quality, despite the large number of questions.

Next, we test whether our 16 heuristics and biases experiments replicate known results at the aggregate level. See Table 2. We see that both in waves 1-3 and in wave 4, all between-subject results replicate those in the literature, with the exception of the base rate fallacy. While Kahneman and Tversky (1973) find that probability assessments are not sensitive to base rate, we find that they are. In terms of within-subject experiments, waves 1-3 and wave 4 also replicate all known results, with the exception of the non-separability of risks and benefits for one of the items, bicycles. While Stanovich and West (2008) find a negative correlation between judged benefits and risks for this item, we find no correlation. The fact that our data replicates the vast majority of these known experimental results is again a sign of good data quality.

Finally, we calculate the test-retest accuracy in our data. We use the answers from waves 1-3 to our 88 holdout questions (across 17 tasks) as the ground truth. Given that all holdout questions are either binary or numerical (or transformed into numerical answers), we calculate accuracy as follows. For binary measures, accuracy is simply a binary indicator of whether two answers match. For non-binary measures, we calculate the absolute deviation between the ground truth and predicted answer, divided by the range of possible answers.[5] We then compute accuracy as 1 minus this absolute deviation. This measure generalizes accuracy from binary to numerical questions: it ranges between 0 and 1, is equal to 1 when the prediction is equal to the ground truth, and 0 when it is maximally different. When multiple questions are included in the same task, we take the mean accuracy across the questions within each task. Therefore, we are left with one measure of accuracy per respondent for each of the 17 tasks (11 between-subject experiments, 5 within-subject experiments, 1 pricing study). Figure 2 reports the average accuracy across respondents for each task as well as 95% confidence interval. We see that the average test-retest accuracy across the 17 tasks is 81.72%. This number is aligned with others reported in the literature (e.g., Park et al. 2024), and again gives us confidence in the data's quality.

## 4 Creation of the digital twins

To construct each digital twin, we begin by merging the original Qualtrics survey files (QSF) with each participant's raw responses, creating a self-contained JSON record for every

---

[5]For the anchoring questions which accept unbounded answers, we transform the data into deciles based on the answers from wave 2 before calculating the absolute deviation.

individual. This record lists, in order, every question the participant actually encountered, the response options shown, and the answers. We then partition this record into three separate files:

- **Persona JSON:** Aggregates all non–hold-out content from waves 1-3, used to define the persona.

- **Evaluation answer-block JSON:** Contains the participant's wave 1–3 responses to hold-out items, providing the ground truth for evaluating simulation accuracy.

- **Retest answer-block JSON:** Stores wave 4 responses to those same hold-out items, used solely to compute the human test–retest accuracy benchmark.

By distributing the data in this modular format, we enable future researchers to experiment with alternative encoding or summarization schemes before presenting the material to an LLM. Future research may also explore alternative ways to split the questions into training and hold-out observations, e.g., predicting the responses to cognitive ability questions based on the answers to the other questions in waves 1-3.

In the present work, we use a straightforward text-based approach: the JSON files are converted into text descriptions detailing the questions, options, and participant answers. The prompt instruction is attached in Web Appendix A2.1. The model's completion is then post-processed back into canonical survey coding and compared with the Wave 1–3 ground truth, yielding the accuracy statistics reported in Figure 2. Our dataset includes all these files for future researchers to train and test LLM-based digital twin simulations.

# 5 Initial tests of digital twins' predictive performance

Section 3 presented evidence that suggests the quality of the data is good, despite the high number of questions. Hence, this dataset should be helpful to researchers and practitioners interested in developing, testing, improving and deploying digital twin simulations. In this section, we present initial tests of the performance of digital twins created using the data, both at the individual and aggregate levels. As mentioned, those initial results may be viewed as providing a lower bound on the predictive performance that may be achieved in the future.

## 5.1 Individual level

To systematically evaluate different strategies for LLM-based persona simulation, we experimented with over a dozen variations in both persona construction and simulation methodology. These include differences in input format (e.g., text vs. JSON), model choices, prompting strategies such as proactive reasoning or chain-of-thought, and persona summaries. Full experimental details are provided in Web Appendix A2.[6] Overall, we find that the predictive accuracy of the answers simulated by the digital twins falls within a similar range across approaches (see Table 1). We hope this collection of baseline results will serve as a useful benchmark for future researchers exploring more advanced methods for persona training, such as reinforcement learning with human feedback (RLHF). For the initial analyses reported here, we focus on the text format using GPT4.1-mini.

Figure 2 reports, for each task, the predictive accuracy of the answers simulated by the digital twins, as well as the accuracy of a random benchmark which chooses each answer from a random uniform distribution. On average, across the 17 tasks the accuracy of the digital twin predictions is 71.72%, and the ratio of the digital twin accuracy to the test-retest accuracy is 87.67%.

---

[6]The appendix also examines benchmarks that require some of the holdout data (split into a training and validation subsample), i.e., fine-tuning and a traditional machine learning benchmark XG Boost.

Table 1: Various persona simulation approaches and evaluation results

| Approach | LLM | Accuracy |
|---|---|---|
| Text Persona | GPT4.1-mini | 71.72% |
| Text Persona | Gemini-flash2.5 | 69.40% |
| JSON Persona | GPT4.1-mini | 70.48% |
| JSON Persona | GPT4.1 | 71.05% |
| Persona Summary | GPT4.1-mini | 68.02% |
| Persona Summary + JSON Persona | GPT4.1-mini | 67.88% |
| Text Persona (Reasoning) | GPT4.1-mini | 70.39% |
| Text Persona (Repeating Questions) | GPT4.1-mini | 70.45% |
| Text Persona (Default Temperature) | GPT4.1-mini | 71.24% |
| JSON Persona (Predicted Output) | GPT4.1-mini | 69.00% |
| JSON Persona (Predicted Output) | GPT4.1 | 71.92% |
| Random Guessing | – | 59.17% |

Figure 2: Predictive accuracy

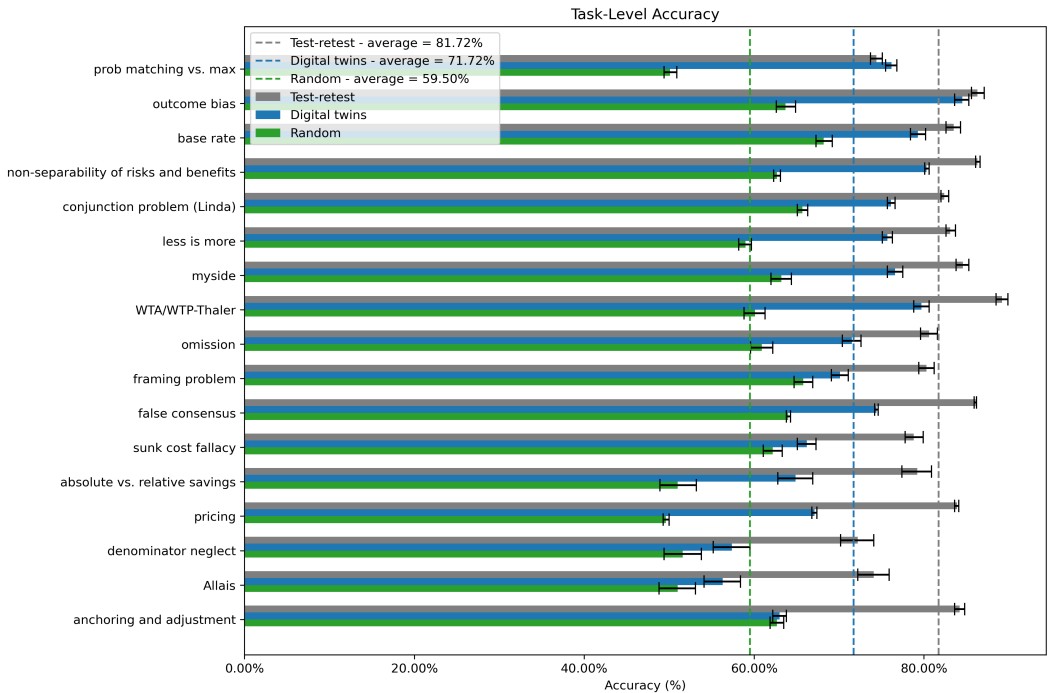

## 5.2 Aggregate level

We test whether the data simulated from the digital twins replicates the average treatment effects from the 11 classic between-subject studies and the 5 classic within-subject studies included in our experiment. Table 2 shows that for 6 of the 10 results replicated by waves 1-3 and wave 4, the results from the digital twins also replicate the results. For anchoring and adjustment, the digital twins replicate the effect when asking participants to estimate the height of the highest redwood tree. But when asking participants to estimate the number of African countries in the UN, 98.8% of the twins gave the correct answer (54) and no anchoring effect was found. In contrast, only about 10% of humans gave the correct answer (8.79% in wave 2, 10.25% in wave 4). Three other between-subject effects were not replicated. In the outcome bias experiment, participants evaluate a physician's decision to operate on a patient. Humans evaluate the decision more favorably when the operation succeeded than when it failed, despite the risk being greater in the first condition. Overall, about 80% of humans gave a favorable evaluation (78.18% in wave 1, 81.39% in wave 4). In contrast,

digital twins all gave a favorable rating ("correct" to "clearly correct"), with no significant different across conditions. In the sunk cost fallacy experiment, the effect was actually reversed with the digital twins vs. their human counterparts, which we hope future research can explore. In the Allais problem experiment, which tests for violation of the independence axiom of utility theory, all digital twins chose the lower risk - lower reward option over the higher risk - higher reward one. Humans, on the other hand, were much more split in their decisions, and showed systematic differences across conditions (which violates the independence axiom of utility theory). Finally, the base rate fallacy, which was replicated neither in wave 2 nor in wave 4, was not replicated by the digital twins either.

Table 2: Replications of heuristics and biases

| Task | Source | Prediction | Replicated Waves 1-3 | Replicated Wave 4 | Twins |
|------|--------|-----------|-------|-------|-------|
| *Between-subject experiments* | | | | | |
| Base rate problem | Kahneman and Tversky (1973) | no difference in prob. assessment when base rate=30 vs. 70 | ✗ | ✗ | ✗ |
| Outcome bias | Baron and Hershey (1988) | average correctness assessment higher in success vs. failure condition | ✓ | ✓ | ✗ |
| Sunk cost fallacy | Stanovich and West (2008) | average number of purchases higher in sunk cost vs. no sunk cost condition | ✓ | ✓ | ✗ |
| Allais problem | Stanovich and West (2008) | violation of independence axiom of utility theory (different choices in Form 1 vs. 2) | ✓ | ✓ | ✗ |
| Framing problem | Tversky and Kahneman (1981) | stronger preference for risky option under loss frame vs. gain frame | ✓ | ✓ | ✓ |
| Conjunction problem (Linda) | Tversky and Kahneman (1983) | probability assessments higher for feminist bank teller vs. bank teller | ✓ | ✓ | ✓ |
| Anchoring and adjustment | Tversky and Kahneman (1974), Epley et al. (2004) | average prediction higher with large vs. small anchor | ✓✓ | ✓✓ | ✓✗ |
| Absolute vs. relative savings | Stanovich and West (2008) | probability of driving to store higher when discount is larger vs. smaller % of price | ✓ | ✓ | ✓ |
| Myside bias | Stanovich and West (2008) | average agreement higher for ban of German car in US vs. American car in Germany | ✓ | ✓ | ✓ |
| Less is More | Stanovich and West (2008) | average attractiveness higher when possibility of loss vs. no possibility of loss | ✓ | ✓ | ✓ |
| WTA/WTP – Thaler problem | Stanovich and West (2008) | WTA-certainty>WTP-certainty>WTP-noncertainty | ✓ | ✓ | ✓ |
| *Within-subject experiments* | | | | | |
| False consensus | Furnas and LaPira (2024) | overpredict (underpredict) public support if own support (oppose) | ✓ | ✓ | ✓ |
| Nonseparability of risk and benefits judgments | Stanovich and West (2008) | negative correlation between benefits and risks for each item | ✓✓✓✗ | ✓✓✓✗ | ✓✗✗✗ |
| Omission bias | Stanovich and West (2008) | significant proportion avoid treatment | ✓ | ✓ | ✗ |
| Probability matching vs. maximizing | Stanovich and West (2008) | significant proportion choose suboptimal strategy | ✓ | ✓ | ✗ |
| Dominator neglect | Stanovich and West (2008) | significant proportion choose non-normative option | ✓ | ✓ | ✓ |

Moving to within-subject experiments, we find that the digital twin results match the human results in two of the five within-subject experiments (see Table 2). In the nonseparability of risk and benefit judgments study, the digital twins judgments display negative correlation as predicted, but the correlation is significant for only one of the items. For probability matching vs. maximizing, the digital twins always selected the normative option while their human counterparts chose the normative option about 30% of the time only. For omission bias, participants were asked whether they would accept a vaccine that prevents catching a flu that has a 10% chance of killing affected patients when the vaccine itself carries a 5% chance of death. While approximately 45% of the human participants (45.10% in wave 2, 44.80% in wave 4) refused the vaccine, only 4.0% of the twins refused the vaccine. This finding echoes our finding related to outcome bias where digital twins were much more favorable to medical professionals compared to their human counterparts.

Finally, we construct average demand curves from the pricing study. Figure A1 shows the average demand curves from the responses from waves 3 vs. wave 4 vs. digital twins. We see that the average demand curves from wave 3 vs. 4 are practically indistinguishable. We find that the average demand curve obtained from the twin is not fully downward sloping, due to the twins' responses to free products. This echoes Gui and Toubia (2023), although digital twins produce demand curves that are downward sloping for positive prices and that are generally closer to the ground truth compared to the demand curves obtained by Gui and Toubia (2023) without such input data.

In sum, while digital twins replicate the majority of between-subject and within-subject effects, there are notable exceptions. Some occur when digital twins fail to mimic the suboptimal or non-normative behaviors of humans, or cannot "unlearn" certain facts (e.g., the number of African countries in the UN). In some areas, twins do match suboptimal human responses (e.g., absolute vs. relative savings, less is more, dominator neglect), raising the broader question of whether digital twins should be seen as "improved" humans or as models that also replicate human deviations from normative behavior and knowledge gaps.

Other deviations appear in the medical domain (e.g., outcome bias, omission bias). Future research should examine whether digital twins are systematically more trusting of the medical profession than humans. Another factor may be that certain topics, such as vaccination, have become highly polarized. This may be considered in light of the result that GPT models tend to struggle to reflect conservative viewpoints (Motoki et al. 2024). For instance, in our false consensus task, while about 45% of humans somewhat or strongly supported increased deportations of those staying in the US illegally (45.42% in wave 1, 45.09% in wave 4), only 25.85% of digital twins did so, and 74.1% strongly or somewhat *opposed* the measure. More research is needed to systematically study where digital twins diverge from humans, especially in medical and political domains, and to identify other domains where such differences may arise.

# 6   Conclusion

We present a unique dataset spanning over 500 questions and 2,000 respondents, with high data quality evidenced by sensible correlation patterns, good test-retest accuracy, and replication of known effects. While this resource can broadly benefit business and social science scholars and practitioners, our primary focus is on using it to build digital twins. In initial tests, these twins predict human behavior with out-of-sample accuracy reaching 88% of the test-retest benchmark. Replication of average treatment effects is generally good, though further research is needed to determine if digital twins can capture non-normative behaviors and reflect the full diversity of political and domain-specific views. We also hope that future research will explore the full range of potential applications of digital twins in marketing, business and beyond. Examples include personalization, targeting, product development, professional development and training, advocacy and negotiations, mental health and counseling, etc. The dataset's focus on the US is a potential limitation. Overall, we hope this resource accelerates LLM research as well as business and social science applications while being mindful of societal risks such as dehumanization of research and excessive reliance on AI in decision-making.

## Funding and Competing Interests Declarations

All authors certify that they have no affiliations with or involvement in any organization or entity with any financial interest or non-financial interest in the subject matter or materials discussed in this manuscript. The research was supported by one funding institution (kept anonymous for the peer review process).

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

## Table 3: Appendix Table: Complete list of questions and related measures

| Task (source) | #Questions (format) | Extracted measure(s) | Wave(s) |
|---|---|---|---|
| *Demographics* | | | |
| Demographics (Santurkar et al. 2023) | 12 (multiple choice) | region, sex, age, education, race, citizenship, marital status, religion, religious attendance, political party, household income, political ideology (categorical) | 1 |
| Additional demographics | 2 (multiple choice) | household size, employment status (categorical) | 1 |
| *Personality Traits* | | | |
| Big 5 personality test (John et al. 1999) | 44 (5-point Likert) | extraversion, agreeableness, conscientiousness, neuroticism, openness scores (numerical) | 1 |
| Need for cognition scale (Cacioppo et al. 1984) | 18 (5-point Likert) | need for cognition score (numerical) | 1 |
| Agentic vs. Communal Values scale (Trapnell and Paulhus 2012) | 24 (9-point Likert) | agency score, communion score (numerical) | 1 |
| Consumer Minimalism scale (Wilson and Bellezza 2022) | 12 (5-point Likert) | minimalism score (numerical) | 1 |
| Empathy scale (Carré et al. 2013) | 20 (5-point Likert) | basic empathy score (numerical) | 1 |
| Green values scale (Haws et al. 2014) | 6 (5-point Likert) | green score (numerical) | 1 |
| Social Desirability scale (Reynolds 1982) | 13 (binary choice) | social desirability score (numerical) | 2 |
| Conscientiousness scale (Johnson et al. 2019) | 8 (9-point Likert) | conscientiousness score (wave 2) (numerical) | 2 |
| Anxiety scale (Beck et al. 1988) | 21 (4-point Likert) | anxiety score (numerical) | 2 |
| Individualism vs. Collectivism scale (Triandis and Gelfand 1998) | 16 (5-point Likert) | horizontal/vertical individualism, horizontal/vertical collectivism scores (numerical) | 2 |
| Selves questionnaire (Higgins et al. 1985) | 3 (open-ended) | n/a | 2 |
| Regulatory Focus scale (Fellner et al. 2007) | 10 (7-point Likert) | regulatory focus score (numerical) | 3 |
| Tightwads vs. Spendthrift scale (Rick et al. 2008) | 4 (multiple choice) | tightwads vs. spendthrift score (numerical) | 3 |
| Depression scale (Date 1987) | 22 (multiple choice) | depression score (numerical) | 3 |
| Need for uniqueness scale (Ruvio et al. 2008) | 12 (5-point Likert) | need for uniqueness score (numerical) | 3 |
| Self-monitoring scale (Lennox and Wolfe 1984) | 13 (6-point Likert) | self-monitoring score (numerical) | 3 |
| Self-concept clarity scale (Campbell et al. 1996) | 12 (5-point Likert) | self-concept clarity score (numerical) | 3 |
| Need for closure scale (Roets and Van Hiel 2011) | 15 (5-point Likert) | need for closure score (numerical) | 3 |
| Maximization scale (Nenkov et al. 2008) | 6 (5-point Likert) | maximization score (numerical) | 3 |
| *Cognitive Abilities* | | | |

| | | | |
|---|---|---|---|
| Cognitive Reflection Test (Krefeld-Schwalb et al. 2024) | 4 (open-ended) | CRT score (numerical) | 1 |
| Fluid intelligence test (Krefeld-Schwalb et al. 2024) | 6 (multiple choice) | fluid intelligence score (numerical) | 1 |
| Crystallized intelligence test (Krefeld-Schwalb et al. 2024) | 20 (multiple choice) | crystallized intelligence score (numerical) | 1 |
| Syllogisms test (Markovits and Nantel 1989) | 12 (multiple choice) | syllogism score (numerical) | 1 |
| Overconfidence (Dean and Ortoleva 2019) | 1 (numerical) | overconfidence score (own predicted-actual score) | 1 |
| Overplacement (Dean and Ortoleva 2019) | 1 (numerical) | overplacement score (own predicted score-predicted average) | 1 |
| Financial literacy test (Johnson et al. 2019) | 7 (mult. choice)+1 (num.) | financial literacy score (numerical) | 2 |
| Numeracy test (Johnson et al. 2019) | 8 (numerical) | numeracy score (numerical) | 2 |
| Deductive certainty of Modus Ponens test (Stanovich and West 2008) | 4 (binary choice) | deductive certainty score | 2 |
| Forward Flow (free associations) (Gray et al. 2019) | 20 (open-ended) | forward flow score (average pairwise semantic distance) | 2 |
| Wason Selection Task (Klauer et al. 2007) | 1 (multiple choice) | Wason Selection Task score (numerical) | 3 |
| *Economic Preferences* | | | |
| Ultimatum game (sender) (Güth et al. 1982) | 1 (multiple choice) | ultimatum-send (percentage sent) | 1 |
| Ultimatum game (receiver) (Güth et al. 1982) | 6 (binary choice) | ultimatum-receive (acceptance probability) | 1 |
| Mental accounting (Thaler 1985) | 4 (binary choice) | mental accounting score (% choices consistent with mental account predictions) | 1 |
| Discount (Dean and Ortoleva 2019) | 3 (multiple price list) | discount rate (numerical) | 2 |
| Present bias (Dean and Ortoleva 2019) | 3 (multiple price list) | present bias (numerical) | 2 |
| Risk Aversion (Dean and Ortoleva 2019) | 3 (uncertainty equivalence) | risk aversion coefficient (numerical) | 2 |
| Loss Aversion (Dean and Ortoleva 2019) | 4 (uncertainty equivalence) | loss aversion coefficient (numerical) | 2 |
| Trust game (sender) (Dean and Ortoleva 2019) | 1 (multiple choice) | trust-send (percentage sent) | 2 |
| Trust game (receiver) (Dean and Ortoleva 2019) | 5 (multiple choice) | Trust-return (average percentage returned) | 2 |
| Trust game (sender) thought listing | 1 (open-ended) | n/a | 2 |
| Trust game (receiver) thought listing | 1 (open-ended) | n/a | 2 |
| Dictator game (Baron and Hershey 1988) | 1 (multiple choice) | dictator-send (percentage sent) | 3 |
| Dictator game thought listing | 1 (open-ended) | n/a | 3 |
| *Heuristics and Biases (between subject)* | | | |

| Base rate problem (Kahneman and Tversky 1973) | 1 (slider scale) | average prob. assessment (numerical) in each condition (base rate of 30 vs. 70 engineers) | 1, 4 |
|---|---|---|---|
| Outcome bias (Baron and Hershey 1988) | 1 (7-point Likert) | average correctness assessment (numerical) in each condition (success vs. failure) | 1, 4 |
| Sunk cost fallacy (Stanovich and West 2008) | 1 (numerical) | average number of purchases (numerical) in each condition (sunk cost yes vs. no) | 1, 4 |
| Allais problem (Stanovich and West 2008) | 1 (binary choice) | lottery choice probability in each condition (form 1 vs. 2) | 1, 4 |
| Framing problem (Tversky and Kahneman 1981) | 1 (6-point Likert) | average preference for B vs. A (numerical) in each condition (framing gain vs. loss) | 2, 4 |
| Conjunction problem (Linda) (Tversky and Kahneman 1983) | 3 (6-point Likert) | average prob. assessment in each condition (feminist bank teller vs. bank teller) | 2,4 |
| Anchoring and adjustment (Tversky and Kahneman 1974, Epley et al. 2004) | 2 (numerical) | average prediction (numerical) in each condition (with high vs. low anchor) | 2, 4 |
| Absolute vs. relative savings (Stanovich and West 2008) | 1 (binary choice) | probability of driving to store in each condition (calculator vs. jacket) | 2,4 |
| Myside bias (Stanovich and West 2008) | 1 (6-point Likert) | average ban agreement (numerical) in each condition (German vs. Ford) | 2,4 |
| Less is More (Stanovich and West 2008) | 3 (5/6-point Likert) | average attractiveness (numerical) in each condition (Form A vs. B vs. C) | 3, 4 |
| WTA/WTP – Thaler problem (Stanovich and West 2008) | 1 (multiple choice) | average in each condition (WTP-certainty, WTA-certainty, WTP-noncertainty) | 3, 4 |
| *Heuristics and Biases (within subject)* | | | |
| False consensus (Furnas and LaPira 2024) | 10 (5-point Likert)+10 (slider) | average predicted public support for each level of own support | 1,4 |
| Nonseparability of risk and benefits judgments (Stanovich and West 2008) | 8 (7-point Likert) | correlation between benefits and risks for each item | 1,4 |
| Omission bias (Stanovich and West 2008) | 1 (4-point Likert) | likelihood of taking vaccine (numerical) | 2,4 |
| Probability matching vs. maximizing (Stanovich and West 2008) | 6-10 (binary choice) | proportion choosing each strategy (Match, Max, other) | 3, 4 |
| Dominator neglect (Stanovich and West 2008) | 1 (binary choice) | proportion choosing large tray | 3,4 |
| *Product Preferences* | | | |
| Pricing study (Gui and Toubia 2023) | 40 (binary choice) | demand curve for each product | 3,4 |

# Appendix

## A1   Additional tables and figures

Table A1: Demographic characteristics of sample

| Category | Count | Percentage |
|---|---|---|
| *Region* | | |
| South | 834 | 40.5% |
| West | 494 | 24.0% |
| Midwest | 372 | 18.1% |
| Northeast | 342 | 16.6% |
| Pacific | 16 | 0.8% |
| *Sex* | | |
| Female | 1044 | 50.7% |
| Male | 1014 | 49.3% |
| *Age* | | |
| 18-29 | 388 | 18.9% |
| 30-49 | 735 | 35.7% |
| 50-64 | 658 | 32.0% |
| 65+ | 277 | 13.5% |
| *Education* | | |
| Less than high school | 17 | 0.8% |
| High school graduate | 272 | 13.2% |
| Some college, no degree | 468 | 22.7% |
| Associate's degree | 253 | 12.3% |
| College graduate/some postgrad | 735 | 35.7% |
| Postgraduate | 313 | 15.2% |
| *Race* | | |
| White | 1361 | 66.1% |
| Black | 251 | 12.2% |
| Hispanic | 194 | 9.4% |
| Asian | 140 | 6.8% |
| Other | 112 | 5.4% |
| *Citizenship* | | |
| Yes | 2054 | 99.8% |
| No | 4 | 0.2% |
| *Marital Status* | | |
| Married | 813 | 39.5% |
| Never been married | 714 | 34.7% |
| Divorced | 218 | 10.6% |
| Living with a partner | 212 | 10.3% |
| Widowed | 70 | 3.4% |
| Separated | 31 | 1.5% |
| *Religion* | | |
| Protestant | 510 | 24.8% |
| Roman Catholic | 358 | 17.4% |
| Nothing in particular | 327 | 15.9% |
| Agnostic | 311 | 15.1% |
| Atheist | 216 | 10.5% |
| Other | 215 | 10.4% |
| Jewish | 39 | 1.9% |
| Buddhist | 25 | 1.2% |
| Muslim | 18 | 0.9% |
| Orthodox | 17 | 0.8% |
| Mormon | 15 | 0.7% |
| Hindu | 7 | 0.3% |
| *Religious Attendance* | | |
| Never | 838 | 40.7% |
| Seldom | 463 | 22.5% |
| Once a week | 295 | 14.3% |
| A few times a year | 246 | 12.0% |
| Once or twice a month | 129 | 6.3% |
| More than once a week | 87 | 4.2% |
| *Political Party* | | |
| Democrat | 847 | 41.2% |
| Independent | 609 | 29.6% |
| Republican | 540 | 26.2% |
| Something else | 62 | 3.0% |
| *Household Income* | | |
| Less than $30,000 | 367 | 17.9% |
| $30,000-$50,000 | 412 | 20.0% |
| $50,000-$75,000 | 411 | 20.0% |
| $75,000-$100,000 | 316 | 15.4% |
| $100,000 or more | 552 | 26.8% |
| *Political Ideology* | | |
| Moderate | 582 | 28.3% |
| Liberal | 564 | 27.4% |
| Conservative | 430 | 20.9% |

| Category | Count | Percentage |
|---|---|---|
| Very liberal | 345 | 16.8% |
| Very conservative | 137 | 6.7% |
| *Household Size* | | |
| 1 | 412 | 20.0% |
| 2 | 650 | 31.6% |
| 3 | 423 | 20.6% |
| 4 | 352 | 17.1% |
| More than 4 | 221 | 10.7% |
| *Employment Status* | | |
| Full-time employment | 871 | 42.3% |
| Self-employed | 280 | 13.6% |
| Part-time employment | 269 | 13.1% |
| Unemployed | 249 | 12.1% |
| Retired | 245 | 11.9% |
| Student | 78 | 3.8% |
| Home-maker | 66 | 3.2% |

Table A2: Summary statistics for individual-level measures

| Measure | Average | Std | Median | Min | Max | Theoretical Range |
|---|---|---|---|---|---|---|
| *Personality Traits* | | | | | | |
| extraversion score | 2.87 | 0.93 | 2.88 | 1 | 5 | [1,5] |
| agreeableness score | 4.00 | 0.69 | 4.00 | 1.22 | 5 | [1,5] |
| conscientiousness score | 3.93 | 0.76 | 4.00 | 1.11 | 5 | [1,5] |
| neuroticism score | 2.71 | 1.00 | 2.63 | 1 | 5 | [1,5] |
| openness score | 3.75 | 0.69 | 3.80 | 1 | 5 | [1,5] |
| need for cognition score | 3.40 | 0.83 | 3.44 | 1 | 5 | [1,5] |
| agency score | 4.99 | 1.36 | 4.83 | 1 | 9 | [1,9] |
| communion score | 6.94 | 1.11 | 7.00 | 1 | 9 | [1,9] |
| minimalism score | 3.44 | 0.78 | 3.50 | 1 | 5 | [1,5] |
| basic empathy score | 3.88 | 0.58 | 3.90 | 1.50 | 5 | [1,5] |
| green score | 3.51 | 1.01 | 3.67 | 1 | 5 | [1,5] |
| social desirability score | 5.71 | 3.74 | 6.00 | 0 | 13 | [0,13] |
| conscientiousness score (wave 2) | 6.40 | 2.12 | 7 | 0 | 8 | {0,...8} |
| anxiety score | 9.84 | 9.60 | 7.00 | 0 | 59 | {0,...63} |
| horizontal individualism score | 4.23 | 0.65 | 4.25 | 1.25 | 5 | [1,5] |
| vertical individualism score | 2.77 | 0.90 | 2.75 | 1 | 5 | [1,5] |
| horizontal collectivism score | 3.90 | 0.73 | 4.00 | 1 | 5 | [1,5] |
| vertical collectivism score | 3.75 | 0.85 | 3.75 | 1 | 5 | [1,5] |
| regulatory focus score | 4.90 | 0.64 | 4.90 | 2.50 | 7 | [1,7] |
| tightwad vs. spendthrift score | 12.72 | 4.56 | 12 | 4 | 24 | {4,...26} |
| depression score | 11.29 | 10.25 | 9 | 0 | 55 | {0,...61} |
| need for uniqueness score | 2.47 | 0.89 | 2.42 | 1 | 5 | [1,5] |
| self-monitoring score | 2.77 | 0.46 | 2.77 | 0.77 | 4.38 | [0,5] |
| self-concept clarity score | 3.60 | 0.97 | 3.75 | 1 | 5 | [1,5] |
| need for closure score | 3.52 | 0.65 | 3.60 | 1 | 5 | [1,5] |
| maximization score | 3.17 | 0.65 | 3.17 | 1 | 5 | [1,5] |
| *Cognitive Abilities* | | | | | | |
| CRT score | 2.03 | 1.23 | 2 | 0 | 4 | {0,...4} |
| fluid intelligence score | 1.60 | 1.36 | 1 | 0 | 4 | {0,...6} |
| crystallized intelligence score | 6.09 | 2.38 | 7 | 0 | 9 | {0,...20} |
| syllogism score | 6.98 | 2.22 | 7 | 1 | 11 | {0,...12} |
| overconfidence score | 12.18 | 7.41 | 13 | -18 | 39 | {-42,...42} |
| overplacement score | 1.99 | 7.93 | 3.00 | -37 | 40 | [-42,42] |
| financial literacy score | 4.99 | 1.36 | 5 | 0 | 7 | {0,...8} |
| numeracy score | 5.43 | 2.09 | 6 | 0 | 8 | {0,...8} |
| deductive certainty score | 3.76 | 0.64 | 4 | 0 | 4 | {0,...4} |
| forward flow score | 0.82 | 0.05 | 0.82 | 0.24 | 0.93 | [0,1] |
| Wason Selection Task score | 2.32 | 0.68 | 2 | 0 | 4 | {0,...4} |
| *Economic Preferences* | | | | | | |
| ultimatum-send | 46.29 | 20.45 | 40.00 | 0 | 100 | [0,100] |

| | | | | | | |
|---|---|---|---|---|---|---|
| ultimatum-receive | 80.32 | 18.04 | 83.33 | 0 | 100 | [0,100] |
| mental accounting score | 72.08 | 24.20 | 75.00 | 0 | 100 | [0,100] |
| discount rate | $9.85 \times 10^{13}$ | $4.47 \times 10^{14}$ | 4.83 | -1 | $4.50 \times 10^{15}$ | $(-1,\infty)$ |
| present bias | 0.04 | 0.11 | 0 | -0.43 | 0.59 | $(-\infty,\infty)$ |
| risk aversion coefficient | 0.12 | 0.24 | 0.07 | -0.67 | 0.83 | $(-\infty,\infty)$ |
| loss aversion coefficient | 0.97 | 0.66 | 0.89 | 0.06 | 6.35 | $[0,\infty)$ |
| trust-send | 48.47 | 31.38 | 40.00 | 0 | 100 | $[0,100]$ |
| trust-return | 40.67 | 17.91 | 45.00 | 0 | 100 | $[0,100]$ |
| dictator-send | 39.10 | 18.85 | 40.00 | 0 | 100 | $[0,100]$ |

Table A3: Examples of significant correlations

| Measure 1 | Measure 2 | Correlation |
|---|---|---|
| need for cognition score | openness score | 0.62 |
| neuroticism score | depression score | 0.62 |
| self-concept clarity score | depression score | -0.55 |
| self-concept clarity score | anxiety score | -0.48 |
| conscientiousness score | depression score | -0.48 |
| agreeableness score | social desirability score | 0.44 |
| neuroticism score | social desirability score | -0.41 |
| conscientiousness score | social desirability score | 0.37 |
| green score | openness score | 0.33 |
| age | self-concept clarity score | 0.33 |

Figure A1: Average demand curves from pricing study

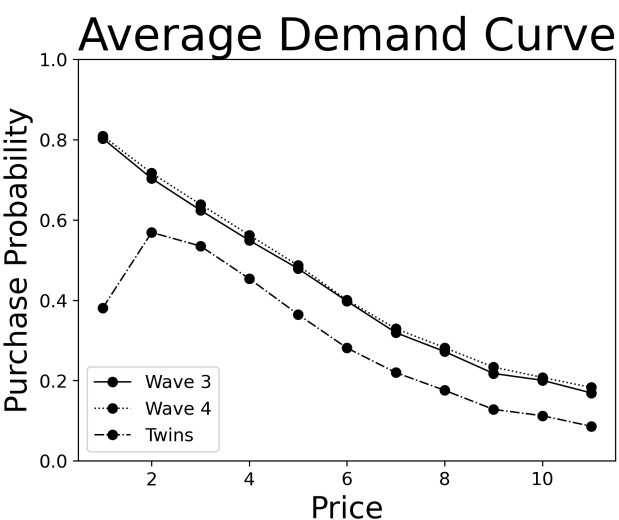

## A2 Various approaches of persona construction/simulation

### A2.1 Approach details

We experimented with a variety of approaches to construct and simulate LLM personas. Below, we describe each approach corresponding to the rows in Table 1.

- **Text Persona & GPT-4.1-mini**: The full set of survey responses is provided as free-form text, and simulation is performed using GPT-4.1-mini.

- **Text Persona & Gemini-flash2.5**: Identical free-text persona input as above, but simulated with Gemini Flash 2.5 to compare model-dependent behavioral fidelity.

- **JSON Persona & GPT-4.1-mini**: Survey responses are encoded as structured JSON fields instead of text, allowing assessment of the impact of input format on model performance.

- **JSON Persona & GPT-4.1**: Same structured JSON input, but using the full GPT-4.1 model to evaluate the effect of increased model capacity on simulation accuracy.

- **Persona Summary & GPT-4.1-mini**: A concise summary of the persona is provided instead of the complete responses, testing model performance under input length constraints.

- **Persona Summary + JSON Persona & GPT-4.1-mini**: The structured JSON persona is augmented with an appended summary to examine whether a hybrid input format improves results.

- **Text Persona (Reasoning) & GPT-4.1-mini**: The text persona is supplemented with explicit instructions for reasoning before providing answers, following a chain-of-thought prompting approach.

- **Text Persona (Repeating Questions) & GPT-4.1-mini**: The model is prompted to repeat each question and answer choice before responding, ensuring full context is processed during simulation.

- **Text Persona (Default Temperature) & GPT-4.1-mini**: Same textual input as the baseline, but with a default sampling temperature (0.7) to evaluate the impact of increased generation randomness (all other conditions use temperature = 0).

- **JSON Persona (Predicted Output) & GPT-4.1-mini**: Utilizes OpenAI's "Predicted Output" feature to test whether efficient and accurate structured output can be generated at lower cost.

- **JSON Persona (Predicted Output) & GPT-4.1**: Same as above, but with the full GPT-4.1 model to examine consistency and potential accuracy gains with a larger model.

- **Random Guessing**: A non-informative baseline in which answers are selected uniformly at random, providing a lower-bound reference for accuracy.

**System prompt:** For all LLM-based simulations, we use the following system prompt:

*You are an AI assistant. Your task is to answer the 'New Survey Question' as if you are the individual described in the 'Persona Profile' (which contains their past survey responses). Remain consistent with the persona's previous answers and stated characteristics. Carefully follow any instructions provided for the new question, including formatting requirements.*

## A2.2 Text Persona vs JSON Persona vs Persona Summary

The **Text Persona** format presents persona information in free-text form, aiming to mimic natural language interaction. An example snippet is shown below:

---

**Text Persona Example**

- **Which part of the United States do you currently live in?**
  *Question Type:* Single Choice
  *Options:*

  1 – Northeast (PA, NY, NJ, RI, CT, MA, VT, NH, ME)

  2 – Midwest (ND, SD, NE, KS, MN, IA, MO, WI, IL, MI, IN, OH)

  3 – South (TX, OK, AR, LA, KY, TN, MS, AL, WV, DC, MD, DE, VA, NC, SC, GA, FL)

  4 – West (WA, OR, ID, MT, WY, CA, NV, UT, CO, AZ, NM)

  5 – Pacific (HI, AK)

  *Answer:* 3 – South (TX, OK, AR, LA, KY, TN, MS, AL, WV, DC, MD, DE, VA, NC, SC, GA, FL)

- **What is the sex that you were assigned at birth?**
  *Question Type:* Single Choice
  *Options:*

  1 – Male

  2 – Female

  *Answer:* 1 – Male

- **How old are you?**
  *Question Type:* Single Choice
  *Options:*

  1 – 18–29

  2 – 30–49

  3 – 50–64

  4 – 65+

  *Answer:* 1 – 18–29

  ...

---

In contrast, the **JSON Persona** format directly feeds the raw JSON structure of the persona into the LLM. A representative snippet is shown below:

---

**JSON Persona Example**

```
[{
  "ElementType": "Block",
  "BlockName": "Demographics",
  "BlockType": "Standard",
  "Questions": [
    {
      "QuestionID": "QID11",
      "QuestionText": "Which part of the United States do you currently live in?",
      "QuestionType": "MC",
      "Options": [
        "Northeast (PA, NY, NJ, RI, CT, MA, VT, NH, ME)",
        "Midwest (ND, SD, NE, KS, MN, IA, MO, WI, IL, MI, IN, OH)",
        "South (TX, OK, AR, LA, KY, TN, MS, AL,
        WV, DC, MD, DE, VA, NC, SC, GA, FL)",
        "West (WA, OR, ID, MT, WY, CA, NV, UT, CO, AZ, NM)",
        "Pacific (HI, AK)"
      ],
      "Settings": {"Selector": "SAVR", "SubSelector": "TX", "ForceResponse": "ON"},
      "Answers": {
```

---

```
        "SelectedByPosition": 3,
        "SelectedText": "South (TX, OK, AR, LA, KY, TN, MS, AL, WV,
        DC, MD, DE, VA, NC, SC, GA, FL)"
      }
    },
    {
      "QuestionID": "QID12",
      "QuestionText": "What is the sex that you were assigned at birth?",
      "QuestionType": "MC",
      "Options": ["Male", "Female"],
      "Settings": {"Selector": "SAVR", "SubSelector": "TX", "ForceResponse": "ON"},
      "Answers": {"SelectedByPosition": 1, "SelectedText": "Male"}
    },
    {
      "QuestionID": "QID13",
      "QuestionText": "How old are you?",
      "QuestionType": "MC",
      "Options": ["18-29", "30-49", "50-64", "65+"],
      "Settings": {"Selector": "SAVR", "SubSelector": "TX", "ForceResponse": "ON"},
      "Answers": {"SelectedByPosition": 1, "SelectedText": "18-29"}
    }
    ...
  ]}
  ...
]
```

In addition, we provide a compressed version of the Text Persona—called the *Persona Summary*—by simplifying the questions and summarizing the responses with distributional information. We envision this format as a more cost-effective option for using personas with LLMs.

### Persona Summary Example

**The person's demographics are as follows:**

- Geographic region: South (TX, OK, AR, LA, KY, TN, MS, AL, WV, DC, MD, DE, VA, NC, SC, GA, FL)
- Gender: Male
- Age: 18–29
- Education level: Some college, no degree
- Race: White
- Citizen of the US: Yes
- Marital status: Never been married
- Religion: Protestant
- Religious attendance: Once or twice a month
- Political affiliation: Republican
- Income: $100,000 or more
- Political views: Conservative
- Household size: 4
- Employment status: Student

**The person's Big Five personality scores are as follows:**

- score_extraversion = 3.5 (75th percentile)
- score_agreeableness = 4.111 (62nd percentile)
- wave1_score_conscientiousness = 4 (53rd percentile)
- score_openness = 3.6 (41st percentile)
- score_neuroticism = 2.5 (47th percentile)
- . . .

In our evaluations, all formats yield comparable accuracy. We provide all formats in the full dataset to maximize flexibility and usability: https://huggingface.co/datasets/LLM-Digital-Twin/Twin-2K-500. The raw JSON format contains the most complete information, with a structured layout that facilitates easy addition, deletion, and retrieval of questions and blocks. The Text Persona format is a human-readable version of the JSON format, designed to resemble natural language prompts compatible with LLM input. The Persona Summary format is a concise alternative to the Text Persona, offering a significantly shorter representation that reduces LLM API usage costs—though at the tradeoff of omitting some detailed information.

### A2.3 Discriminative machine learning benchmark and fine tuning

The benchmarks reported above do not use any of the holdout data. Here we examine benchmarks that rely on splitting the holdout data between training and validation samples. These benchmarks represents a fundamentally different task from LLM simulation: while LLM-based digital twins face an out-of-distribution challenge that requires predicting a person's responses to holdout questions without leveraging any answers from those holdout questions, the benchmarks in this subsection perform an out-of-sample but in-distribution prediction task—they learn from thousands of examples where they observe how other participants' non-holdout responses relate to their holdout responses. However, these benchmarks are quite useful for evaluating the relevance of the questions we collected and also contextualizing the performance of the digital twin simulation: if these questions were not predictive of individual behaviors even in the presence of actual labeled data, it would be difficult to expect that LLM digital twins could simulate realistic behavior in the absence of such labeled data.

- **XG Boost**: We consider a machine learning benchmark that uses XGBoost to predict responses to holdout questions using numeric non-holdout questions from waves 1-3 through cross-fitting. We implement nested cross-validation where the outer loop uses 5 folds, with each fold serving as a held-out test set once. For each outer training set containing four-fifths of the data, we run an inner 3-fold cross-validation with 30-trial randomized hyperparameter search and early stopping. The best hyperparameters from the inner loop are then used to retrain on the full outer training set before making predictions on the held-out fold. For each of the holdout questions, we train a separate XGBoost model using participants' responses to all numeric non-holdout questions (demographics, personality scores, cognitive ability scores, economic preference measures) as features. Each model uses approximately 1,600 training examples per fold. XGBoost achieves 75.7% accuracy, compared to 71.72% for the base LLM approach and 81.72% for the test-retest benchmark. This indicates that the non-holdout questions contain predictive information about behavioral responses, reaching 92.6% of the test-retest benchmark. The relatively small gap between XGBoost (75.7%) and LLM digital twins (71.72%) is encouraging, suggesting that digital twins can capture meaningful variation in behavioral responses even without access to holdout question labels.

- **Fine-tuning**: Another approach is to fine-tune the LLM directly on the training data. In our initial trial, we applied supervised fine-tuning (SFT) using OpenAI's Fine-tuning API with default parameters. The model (GPT-4.1-mini) is lightly fine-tuned on 500 labeled personas. The resulting model achieved 69.61% accuracy, which is actually lower than the original, non-fine-tuned version. This highlights the well-known technical challenges of fine-tuning LLMs. A more rigorous investigation of fine-tuning methods, including both SFT and reinforcement learning, is left for future work.

## A3 Detailed instruments and measures

### A3.1 Demographic variables

*Which part of the United States do you currently live in?* [Northeast (PA, NY, NJ, RI, CT, MA, VT, NH, ME); Midwest (ND, SD, NE, KS, MN, IA, MO, WI, IL, MI, IN, OH); South (TX, OK, AR, LA, KY, TN, MS, AL, WV, DC, MD, DE, VA, NC, SC, GA, FL); West (WA, OR, ID, MT, WY, CA, NV, UT, CO, AZ, NM); Pacific (HI, AK)]

*What is the sex that you were assigned at birth?* [Male; Female]

*How old are you?* [18-29; 30-49; 50-64; 65+]

*What is the highest level of schooling or degree that you have completed?* [Less than high school; High school graduate; Some college, no degree; Associate's degree; College graduate/some postgrad; Postgraduate]

*What is your race or origin?* [White; Black; Asian; Hispanic; Other]

*Are you a citizen of the United States?* [Yes; No]

*Which of these best describes you?* [Married; Living with a partner; Divorced; Separated; Widowed; Never been married]

*What is your present religion, if any?* [Protestant; Roman Catholic; Mormon; Orthodox; Jewish; Muslim; Buddhist; Hindu; Atheist; Agnostic; Other; Nothing in particular]

*Aside from weddings and funerals, how often do you attend religious services?* [More than once a week; Once a week; Once or twice a month; A few times a year; Seldom; Never]

*In politics today, do you consider yourself a* [Republican; Democrat; Independent; Something else]

*Last year, what was your total family income from all sources, before taxes?* [Less than $30,000; $30,000-$50,000; $50,000-$75,000; $75,000-$100,000; $100,000 or more]

*In general, would you describe your political views as* [Very conservative; Conservative; Moderate; Liberal; Very liberal]

*Including yourself, how many people currently live in your household?* [1; 2; 3; 4; More than 4]

*What is your current employment status?* [Full-time employment; Part-time employment; Unemployed; Self-employed; Home-maker; Student; Retired]

### A3.2 Personality traits

#### A3.2.1 Big 5 Personality Test (*John et al. 1999*)

*Here are a number of characteristics that may or may not apply to you. Please indicate next to each statement the extent to which you agree or disagree with that statement. I see myself as someone who...*

Response scale: Disagree strongly (1); Disagree a little (2); Neither agree nor disagree (3); Agree a little (4); Agree strongly (5)

Items: Is talkative (1); Tends to find fault with others (2); Does a thorough job (3); Is depressed, blue (4); Is original, comes up with new ideas (5); Is reserved (6); Is helpful and unselfish with others (7); Can be somewhat careless (8); Is relaxed, handles stress well (9); Is curious about many different things (10); Is full of energy (11); Starts quarrels with others (12); Is a reliable worker (13); Can be tense (14); Is ingenious, a deep thinker (15); Generates a lot of enthusiasm (16); Has a forgiving nature (17); Tends to be disorganized (18); Worries a lot (19); Has an active imagination (20); Tends to be quiet (21); Is generally trusting (22); Tends to be lazy (23); Is emotionally stable, not easily upset (24); Is inventive (25); Has an assertive personality (26); Can be cold and aloof (27); Perseveres until the task is finished (28); Can be moody (29); Values artistic, aesthetic experiences (30); Is sometimes shy, inhibited (31); Is considerate and kind to almost everyone (32); Does things efficiently (33); Remains calm in tense situations (34); Prefers work that is routine (35); Is outgoing, sociable (36); Is sometimes rude to others (37); Makes plans and follows through with them (38); Gets nervous easily (39); Likes to reflect, play with ideas (40); Has few artistic interests (41); Likes to cooperate with others (42); Is easily distracted (43); Is sophisticated in art, music, or literature (44).

Scores: extraversion: 1, 6R, 11, 16, 21R, 26, 31R, 36; agreeableness: 2R, 7, 12R, 17, 22, 27R, 32,

37R, 42; conscientiousness: 3, 8R, 13, 18R, 23R, 28, 33, 38, 43R; openness: 5, 10, 15, 20, 25, 30, 35R, 40, 41R, 44; neuroticism: 4, 9R, 14, 19, 24R, 29, 34R, 39.

### A3.2.2 Need for cognition scale (Cacioppo et al. 1984)

*Here are a number of characteristics that may or may not apply to you. Please indicate next to each statement the extent to which you agree or disagree with that statement.*
Response scale: Disagree strongly (1); Disagree a little (2); Neither agree nor disagree (3); Agree a little (4); Agree strongly (5)
Items: I would prefer complex to simple problems; I like to have the responsibility of handling a situation that requires a lot of thinking; Thinking is not my idea of fun *; I would rather do something that requires little thought than something that is sure to challenge my thinking abilities *; I try to anticipate and avoid situations where there is likely chance I will have to think in depth about something *; I find satisfaction in deliberating hard and for long hours ; I only think as hard as I have to *; I prefer to think about small, daily projects to long-term ones *; I like tasks that require little thought once I've learned them *; The idea of relying on thought to make my way to the top appeals to me; I really enjoy a task that involves coming up with new solutions to problems; Learning new ways to think doesn't excite me very much *; I prefer my life to be filled with puzzles that I must solve; The notion of thinking abstractly appeals to me; I would prefer a task that is intellectual, difficult, and important to one that is somewhat important but does not require too much thought; I feel relief rather than satisfaction after completing a task that requires a lot of mental effort; It's enough for me that something gets the job done; I don't care how or why it works *; I usually end up deliberating about issues even when they do not affect me personally.
Score: need for cognition. *: reverse-coded items.

### A3.2.3 Agentic vs. Communal Values scale (Trapnell and Paulhus 2012)

*Below are 24 different values that people rate of different importance in their lives. FIRST READ THROUGH THE LIST to familiarize yourself with all the values. While reading over the list, consider which ones tend to be most important to you and which tend to be least important to you. After familiarizing yourself with the list, rate the relative importance of each value to you as "A GUIDING PRINCIPLE IN MY LIFE." It is important to spread your ratings out as best you can—be sure to use some numbers in the lower range, some in the middle range, and some in the higher range. Avoid using too many similar numbers. Work fairly quickly.*
Response scale: Not Important to me 1: 1; 2; 3; 4; Quite Important to me 5: 5; 6; 7; 8; Highly Important to me 9: 9
Items: WEALTH (financially successful, prosperous) (1); PLEASURE (having one's fill of life's pleasures and enjoyments) (2); FORGIVENESS (pardoning others' faults, being merciful) (3); INFLUENCE (having impact, influencing people and events) (4); TRUST (being true to one's word, assuming good in others) (5); COMPETENCE (displaying mastery, being capable, effective) (6); HUMILITY (appreciating others, being modest about oneself) (7); ACHIEVEMENT (reaching lofty goals) (8); ALTRUISM (helping others in need) (9); AMBITION (high aspirations, seizing opportunities) (10); LOYALTY (being faithful to friends, family, and group) (11); POLITENESS (courtesy, good manners) (12); POWER (control over others, dominance) (13); HARMONY (good relations, balance, wholeness) (14); EXCITEMENT (seeking adventure, risk, an exciting lifestyle) (15); HONESTY (being genuine, sincere) (16); COMPASSION (caring for others, displaying kindness) (17); STATUS (high rank, wide respect) (18); CIVILITY (being considerate and respectful toward others) (19); AUTONOMY (independent, free of others' control) (20); EQUALITY (human rights and equal opportunity for all) (21); RECOGNITION (becoming notable, famous, or admired) (22); TRADITION (showing respect for family and cultural values) (23); SUPERIORITY (defeating the competition, standing on top) (24).
Scores: agency: 1, 2, 4, 6, 8, 10, 13, 15, 18, 20, 22, 24; communion: 3, 5, 7, 9, 11, 12, 14, 16, 17, 19, 21, 23.

*A3.2.4   Consumer Minimalism scale (Wilson and Bellezza 2022)*

627

628 *Please indicate your agreement with each of the following statements about yourself.*
629 Response scale: Disagree strongly (1); Disagree a little (2); Neither agree nor disagree (3);
630 Agree a little (4); Agree strongly (5)
631 Items: I avoid accumulating lots of stuff (1); I restrict the number of things I own (2); "Less
632 is more" when it comes to owning things (3); I actively avoid acquiring excess possessions
633 (4); I am drawn to visually sparse environments (5); I prefer simplicity in design (6); I keep
634 the aesthetic in my home very sparse (7); I prefer leaving spaces visually empty over filling
635 them (8); I am mindful of what I own (9); The selection of things I own has been carefully
636 curated (10); It is important to me to be thoughtful about what I choose to own (11); My
637 belongings are mindfully selected (12).
638 Score: consumer minimalism (average of all items).

639 *A3.2.5   Empathy scale (Carré et al. 2013)*

640

641 *Please indicate your agreement with each of the following statements about yourself.*
642 Response scale: Disagree strongly (1); Disagree a little (2); Neither agree nor disagree (3);
643 Agree a little (4); Agree strongly (5)
644 Items: My friends' emotions don't affect me much *; After being with a friend who is sad
645 about something, I usually feel sad.; I can understand my friend's happiness when she/he
646 does well at something.; I get frightened when I watch characters in a good scary movie.; I
647 get caught up in other people's feelings easily.; I find it hard to know when my friends are
648 frightened. *; I don't become sad when I see other people crying. *; Other people's feelings
649 don't bother me at all. *; When someone is feeling down I can usually understand how
650 they feel.; I can usually work out when my friends are scared.; I often become sad when
651 watching sad things on TV or in films.; I can often understand how people are feeling even
652 before they tell me.; Seeing a person who has been angered has no effect on my feelings. *; I
653 can usually work out when people are cheerful.; I tend to feel scared when I am with friends
654 who are afraid.; I can usually realize quickly when a friend is angry.; I often get swept up in
655 my friends' feelings.; My friend's unhappiness doesn't make me feel anything. *; I am not
656 usually aware of my friends' feelings. *; I have trouble figuring out when my friends are
657 happy. *
658 Score: basic empathy. *: reverse-coded items.

659 *A3.2.6   Green values scale (Haws et al. 2014)*

660

661 *Here are a number of characteristics that may or may not apply to you. Please indicate next to each*
662 *statement the extent to which you agree or disagree with that statement.*
663 Response scale: Disagree strongly (1); Disagree a little (2); Neither agree nor disagree (3);
664 Agree a little (4); Agree strongly (5)
665 Items: It is important to me that the products I use do not harm the environment.; I consider
666 the potential environmental impact of my actions when making many of my decisions.;
667 My purchase habits are affected by my concern for our environment.; I am concerned
668 about wasting the resources of our planet.; I would describe myself as environmentally
669 responsible.; I am willing to be inconvenienced in order to take actions that are more
670 environmentally friendly.
671 Score: green (average of all items).

672 *A3.2.7   Social Desirability scale (Reynolds 1982)*

673

674 *Listed below are a number of statements concerning personal attributes and traits. Read each item*
675 *and decide whether the statement is true or false as it pertains to your personally.*
676 Response scale: FALSE, TRUE
677 Items: It is sometimes hard for me to go on with my work if I am not encouraged (1); I

sometimes feel resentful when I don't get my way (2); On a few occasions, I have given up doing something because I thought too little of my ability (3); There have been times when I felt like rebelling against people in authority even though I knew they were right (4); No matter who I'm talking to, I'm always a good listener (5); There have been occasions when I took advantage of someone (6); I'm always willing to admit when I make a mistake (7); I sometimes try to get even rather than forgive and forget (8); I am always courteous, even to people who are disagreeable (9); I have never been irked when people expressed ideas different from my own (10); There have been times when I was quite jealous of the good fortune of others (11); I am sometimes irritated by people who ask favors of me (12); I have never deliberately said something that hurt someone's feelings (13).
Score: social desirability (sum of TRUE responses to items 5, 7, 9, 10, 13 and FALSE responses to items 1, 2, 3, 4, 6, 8, 11, 12).

### A3.2.8   Conscientiousness scale (*Johnson et al. 2019*)

*Following are a number of characteristics that may or may not apply to you. Please indicate next to each statement the extent to which that statement accurately or inaccurately describes you.*
Response scale: Extremely inaccurate: 1; 2; 3; 4; Neither inaccurate nor accurate: 5; 6; 7; 8: Extremely accurate: 9
Items: Organized (1); Efficient (2); Systematic (3); Practical (4); Disorganized (5); Sloppy (6); Inefficient (7); Careless (8)
Score: conscientiousness scale (wave 2). number of items 1-4 for which response $> 5$ plus items 5-8 for which response $< 5$.

### A3.2.9   Anxiety scale (*Beck et al. 1988*)

*How much have you been bothered by each of the following symptoms over the past week?*
Response scale: Not at all: 0; 1; 2; Severely - I barely could stand it: 3
Items: Numbness or tingling; Feeling hot; Wobbliness in legs; Unable to relax; Fear of the worst happening; Dizzy or lightheaded; Unsteady; Terrified; Nervous; Feeling of choking; Hands trembling; Shaky; Fear of losing control; Difficulty breathing; Fear of dying; Scared; Indigestion or discomfort in abdomen; Faint; Face flushed; Sweating (not due to heat); Heart pounding or racing.
Score: anxiety score (add up numerical values across items).

### A3.2.10   Individualism vs. Collectivism scale (*Triandis and Gelfand 1998*)

*Following are a number of characteristics that may or may not apply to you. Please indicate next to each statement the extent to which you agree or disagree with that statement.*
Response scale: Disagree strongly (1); Disagree a little (2); Neither agree nor disagree (3); Agree a little (4); Agree strongly (5)
Items: I'd rather depend on myself than others (1); I rely on myself most of the time, I rarely rely on others (2); I often do my own thing (3); My personal identity, independent of others, is very important to me (4); It is important for me to do my job better than the others (5); Winning is everything (6); Competition is the law of nature (7); When another person does better than I do, I get tense and aroused (8); If a co-worker gets a prize, I would feel proud (9); The well-being of my coworkers is important to me (10); To me, pleasure is spending time with others (11); I feel good when I cooperate with others (12); Parents and children must stay together as much as possible (13); It is my duty to take care of my family, even when I have to sacrifice what I want (14); Family members should stick together, no matter what sacrifices are required (15); It is important to me that I respect the decision made by my groups (16).
Scores: horizontal individualism (items 1-4), vertical individualism (items 5-8), horizontal collectivism (items 9-12), vertical collectivism (items 13-16).

### A3.2.11 Selves questionnaire ([Higgins et al. 1985](#))

*In this task we would like you to write about the type of person you aspire to be vs. the person you ought to be vs. the person you actually are.*
*1. Please describe the type of person you aspire to be. That is, write about the traits and behaviors you would like ideally to possess, your ultimate goals for yourself. Please write at least 3 sentences.*
*2. Please describe the type of person you ought to be. That is, write about the traits and behaviors attributes that you should or ought to possess, based on your responsibilities and what other people expect from you. Please write at least 3 sentences.*
*3. Please describe the type of person you actually are. That is, write about the traits and behaviors you actually possess. Please write at least 3 sentences.*
Response format: one text box per question.

### A3.2.12 Regulatory Focus scale ([Fellner et al. 2007](#))

*Here are a number of characteristics that may or may not apply to you. Please indicate next to each statement the extent to which it is true or untrue.*
Response scale: Disagree untrue (1); Not true (2); Probably not true (3); Neither true nor untrue (4); Probably true (5); True (6); Definitely true (7)
Items: I prefer to work without instructions from others; Rules and regulations are helpful and necessary for me; For me, it is very important to carry out the obligations placed on me; I generally solve problems creatively; I'm not bothered about reviewing or checking things really closely; I like to do things in a new way; I always try to make my work as accurate and error-free as possible; I like trying out lots of different things, and am often successful in doing so; It is important to me that my achievements are recognized and valued by other people; I often think about what other people expect of me.
Scores: regulatory focus score (average numerical values across items).

### A3.2.13 Tightwads vs. Spendthrift scale ([Rick et al. 2008](#))

Question 1: *Which of the following best describes your spending habits?*
Response scale: Tightwad (difficulty spending money): 1; 2; 3; 4; 5; About the same or neither: 6; 7; 8; 9; 10; Spendthrift (difficulty controlling spending): 11

Question 2a: *Some people have trouble limiting their spending: they often spend money -*
*for example on clothes, meals, vacations - when they would do better not to.*
*How well does the first description fit you? That is, do you have trouble limiting your spending?*
Response scale: Never (1); Rarely (2); Sometimes (3); Often (4); Always (5)

Question 2b: *Other people have trouble spending money. Perhaps because spending money makes them anxious, they often don't spend money on things they should spend it on.*
*How well does the second description fit you? That is, do you have trouble spending money?*
Response scale: Never (1); Rarely (2); Sometimes (3); Often (4); Always (5)

Question 3: *Following is a scenario describing the behavior of two shoppers. After reading about each shopper, please answer the question that follows.*
*Mr. A is accompanying a good friend who is on a shopping spree at a local mall. When they enter a large department store, Mr. A sees that the store has a "one-day-only-sale" where everything is priced 10-60% off. He realizes that he doesn't need anything, yet can't resist and ends up spending almost $100 on stuff.*
*Mr. B is accompanying a good friend who is on a shopping spree at a local mall. When they enter a large department store, Mr. B sees that the store has a "one-day-only-sale" where everything is priced 10-60% off. He figures that he can get great deals on many items that he needs, yet the thought of spending the money keeps him from buying the stuff.*
*In terms of your own behavior, who are you more similar to, Mr. A or Mr. B?*
Response scale: Mr. A: 1; 2; About the same or neither: 3; 4; Mr. B: 5

Scores: tightwads vs. spendthrift score (add up numerical values across items, questions 2b and 3 are reverse coded).

### A3.2.14    Depression scale (Date 1987)

*This page contains groups of statements. After reading each group of statements carefully, choose the one statement which best describes the way you have been feeling in the past week, including today. If several statements within a group seem to apply equally well, select each one. Be sure to read all the statements in each group before making your choice.*

Items:

1. I do not feel sad (0); I feel sad (1); I am sad all the time and I can't snap out of it (2); I am so sad or unhappy that I can't stand it (3)

2. I am not particularly discouraged about the future (0); I feel discouraged about the future (1); I feel that I have nothing to look forward to (2); I feel that the future is hopeless and that things cannot improve (3)

3. I do not feel like a failure (0); I feel that I have failed more than the average person (1); As I look back on my life, all I can see is a lot of failures (2); I feel I am a complete failure as a person (3)

4. I get as much satisfaction out of things as I used to (0); I don't enjoy things the way I used to (1); I don't get real satisfaction out of anything anymore (2); I am dissatisfied or bored with everything (3)

5. I don't feel particularly guilty (0); I feel guilty a good part of the time (1); I feel guilty most of the time (2); I feel guilty all the time (3)

6. I don't feel I am being punished (0); I feel I may be punished (1); I expect to be punished (2); I feel I am being punished (3)

7. I don't feel disappointed in myself (0); I feel disappointed in myself (1); I am disgusted with myself (2); I hate myself (3)

8. I don't feel I am worse than anybody else (0); I am critical of myself for my weaknesses or mistakes (1); I blame myself all the time for my faults (2); I blame myself for everything bad that happens (3)[7]

10. I don't cry any more than usual (0); I cry more than I used to (1); I cry all the time now (2); I used to be able to cry, but now I can't cry even though I want to (3)

11. I am no more irritated now than I ever am (0); I get annoyed or irritated more easily than I used to (1); I feel irritated all the time now (2); I don't get irritated at all by the things that used to irritate me (3)

12. I have not lost interest in other people (0); I am less interested in other people than I used to be (1); I have lost most of my interest in other people (2); I have lost all of my interest in other people (3)

13. I make decisions about as well as I ever could (0); I put off making decisions more than I used to (1); I have greater difficulty in making decisions than before (2); I can't make decisions at all anymore (3)

14. I do not feel that I am worthless (0); I don't consider myself as worthwhile and useful as I used to (1); I feel more worthless as compared to other people (2); I feel utterly worthless (3)

15. I can work about as well as before (0); It takes an extra effort to get started at doing something (1); I have to push myself very hard to do anything (2); I can't do any work at all (2)

16. I can sleep as well as usual (0); I don't sleep as well as usual (1); I wake up 1-2 hours earlier than usual and find it hard to get back to sleep (2); I wake up several hours earlier than I used to and cannot get back to sleep (3)

17. I don't get more tired than usual (0); I get tired more easily than I used to (1); I get tired from doing almost anything (2); I am too tired to do anything (3)

18. My appetite is no worse than usual (0); My appetite is not as good as it used to be (1); My appetite is much worse now (2); I have no appetite at all anymore (3)

19. I haven't lost much weight, if any, lately (0); I have lost more than 5 pounds (1); I have

---

[7]Item 9, which is about suicidal thoughts, was skipped.

lost more than 10 pounds (2); I have lost more than 15 pounds (3)

20. I am no more worried about my health than usual (0); I am worried about physical problems such as aches and pains; or upset stomach; or constipation (1); I am very worried about physical problems and it's hard to think of much else (2); I am so worried about my physical problems that I cannot think about anything else

21. I have not noticed any recent change in my interest in sex (0); I am less interested in sex than I used to be (1); I am much less interested in sex now (2); I have lost interest in sex completely (3)

22. I am purposely trying to lose weight by eating less. Yes (1); No (0)

Score: depression score (add up numerical values across items).

### A3.2.15   Need for uniqueness scale (*Ruvio et al. 2008*)

*Here are a number of characteristics that may or may not apply to you. Please indicate next to each statement the extent to which you agree or disagree with that statement.*

Response scale: Disagree strongly (1); Disagree a little (2); Neither agree nor disagree (3); Agree a little (4); Agree strongly (5)

Items: I often combine possessions in such a way that I create a personal image that cannot be duplicated; I often try to find a more interesting version of run-of-the-mill products because I enjoy being original; I actively seek to develop my personal uniqueness by buying special products or brands; Having an eye for products that are interesting and unusual assists me in establishing a distinctive image; When it comes to the products I buy and the situations in which I use them, I have broken customs and rules; I have often violated the understood rules of my social group regarding what to buy or own; I have often gone against the understood rules of my social group regarding when and how certain products are properly used.; I enjoy challenging the prevailing taste of people I know by buying something they would not seem to accept.; When a product I own becomes popular among the general population, I begin to use it less.; I often try to avoid products or brands that I know are bought by the general population.; As a rule, I dislike products or brands that are customarily bought by everyone.; The more commonplace a product or brand is among the general population, the less interested I am in buying it.

Score: need for uniqueness score (average numerical values across items).

### A3.2.16   Self-monitoring scale (*Lennox and Wolfe 1984*)

*Here are a number of characteristics that may or may not apply to you. Please indicate next to each statement the extent to which that statement is true or false.*

Response scale: Certainly, always false (0); Generally false (1); Somewhat false, but with exceptions (2); Somewhat true, but with exceptions (3); Generally true (4); Certainly, always true (5)

Items: In social situations, I have the ability to alter my behavior if I feel that something else is called for.; I have the ability to control the way I come across to people, depending on the impression I wish to give them.; When I feel that the image I am portraying isn't working, I can readily change it to something that does.; I have trouble changing my behavior to suit different people and different situations. *; I have found that I can adjust my behavior to meet the requirements of any situation I find myself in.; Even when it might be to my advantage, I have difficulty putting up a good front. *; Once I know what the situation calls for, it's easy for me to regulate my actions accordingly.; I am often able to read people's true emotions correctly through their eyes.; In conversations, I am sensitive to even the slightest change in the facial expression of the person I'm conversing with.; My powers of intuition are quite good when it comes to understanding others' emotions and motives.; I can usually tell when others consider a joke to be in bad taste, even though they may laugh convincingly.; I can usually tell when I've said something inappropriate by reading it in the listener's eyes.; If someone is lying to me, I usually know it at once from that person's manner of expression.

Score: self-monitoring score (average numerical values across items). *: reverse-scored items.

### A3.2.17 Self-concept clarity scale (*Campbell et al. 1996*)

*Please indicate your agreement with each of the following statements about yourself.*
Response scale: Disagree strongly (1); Disagree a little (2); Neither agree nor disagree (3);
Agree a little (4); Agree strongly (5)
Items: My beliefs about myself often conflict with one another. *; On one day I might have
one opinion of myself and on another day I might have a different opinion. *; I spend a lot
of time wondering about what kind of person I really am. *; Sometimes I feel that I am not
really the person that I appear to be. *; When I think about the kind of person I have been
in the past, I'm not sure what I was really like. *; I seldom experience conflict between the
different aspects of my personality.; Sometimes I think I know other people better than I
know myself. *; My beliefs about myself seem to change very frequently. *; If I were asked
to describe my personality, my description might end up being different from one day to
another day. *; Even if I wanted to, I don't think I would tell someone what I'm really like. *;
In general, I have a clear sense of who I am and what I am.; It is often hard for me to make
up my mind about things because I don't really know what I want. *
Score: self-concept clarity score. *: reverse-coded items.

### A3.2.18 Need for closure scale (*Roets and Van Hiel 2011*)

*Please indicate your agreement with each of the following statements about yourself.*
Response scale: Disagree strongly (1); Disagree a little (2); Neither agree nor disagree (3);
Agree a little (4); Agree strongly (5)
Items: I don't like situations that are uncertain.; I dislike questions which could be answered
in many different ways.; I find that a well ordered life with regular hours suits my temper-
ament.; I feel uncomfortable when I don't understand the reason why an event occurred
in my life.; I feel irritated when one person disagrees with what everyone else in a group
believes.; I don't like to go into a situation without knowing what I can expect from it.;
When I have made a decision, I feel relieved.; When I am confronted with a problem, I'm
dying to reach a solution very quickly.; I would quickly become impatient and irritated if
I would not find a solution to a problem immediately.; I don't like to be with people who
are capable of unexpected actions.; I dislike it when a person's statement could mean many
different things.; I find that establishing a consistent routine enables me to enjoy life more.; I
enjoy having a clear and structured mode of life.; I do not usually consult many different
opinions before forming my own view.; I dislike unpredictable situations.
Score: need for closure score (average numerical values across items).

### A3.2.19 Maximization scale (*Nenkov et al. 2008*)

*Please indicate your agreement with each of the following statements about yourself.*
Response scale: Disagree strongly (1); Disagree a little (2); Neither agree nor disagree (3);
Agree a little (4); Agree strongly (5)
Items: When I am in the car listening to the radio, I often check other stations to see if
something better is playing, even if I am relatively satisfied with what I'm listening to.; No
matter how satisfied I am with my job, it's only right for me to be on the lookout for better
opportunities.; I often find it difficult to shop for a gift for a friend.; When shopping, I have
a hard time finding clothing that I really love.; No matter what I do, I have the highest
standards for myself.; I never settle for second best.
Score: maximization score (average numerical values across items).

## A3.3 Cognitive abilities

### A3.3.1 Cognitive Reflection Test (*Krefeld-Schwalb et al. 2024*)

Questions (response format, correct answer):

*Emily's father has three daughters. The first two are named April and May. What is the third daughter's name?* (text, Emily)

*How many cubic feet of dirt are there in a hole that is 3' deep x 3' wide x 3' long? Enter a number of cubic feet.* (non-negative number, 0)

*If you're running a race and you pass the person in second place, what place are you in? Enter a number.* (non-negative number, 2)

*A farmer had 15 sheep and all but 8 died. How many are left? Enter a number* (non-negative number, 8)

Score: CRT (number of correct responses).

### A3.3.2   Fluid intelligence test ([Krefeld-Schwalb et al. 2024](#))

Questions (response format, correct answer):

*Please indicate which is the best answer to complete the figure below*

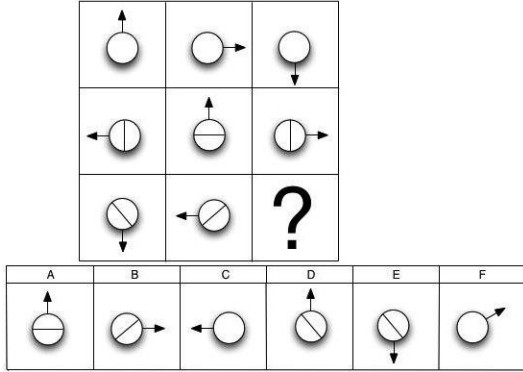

(Multiple choice, D)

(When creating digital twins this image was replaced with the following text: *Puzzle Grid (3x3 Matrix). Each cell in a 3x3 matrix contains a circle with an arrow pointing in one of four directions (up, down, left, right) and sometimes includes a diagonal or horizontal/vertical line inside the circle. We'll label the grid as follows: Top row (Row 1): Cell 1-1: Plain circle with arrow pointing up. Cell 1-2: Plain circle with arrow pointing right. Cell 1-3: Plain circle with arrow pointing down. Middle row (Row 2): Cell 2-1: Circle with a horizontal line, arrow pointing left. Cell 2-2: Circle with a horizontal line, arrow pointing up. Cell 2-3: Circle with a horizontal line, arrow pointing right. Bottom row (Row 3): Cell 3-1: Circle with a diagonal line from top-left to bottom-right, arrow pointing down. Cell 3-2: Circle with a diagonal line from top-left to bottom-right, arrow pointing left. Cell 3-3: Missing (marked with a question mark – this is what we are trying to determine). Answer Choices (Labeled A to F): Each option consists of a circle, possibly with an internal line, and an arrow in a particular direction. A: Circle with horizontal line, arrow pointing up. B: Circle with diagonal line (top-left to bottom-right), arrow pointing left. C: Plain circle, arrow pointing left. D: Circle with diagonal line (top-left to bottom-right), arrow pointing right. E: Circle with diagonal line (top-left to bottom-right), arrow pointing down. F: Plain circle, arrow pointing up-right (diagonal).*
The 5 other images in this section replaced similarly)

*Please indicate which is the best answer to complete the figure below*

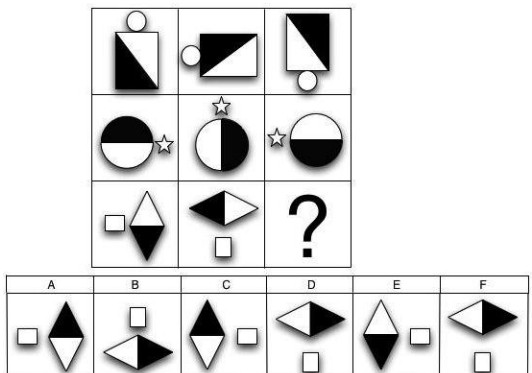

977

(Multiple choice, C)

978

979

980 *Please indicate which is the best answer to complete the figure below*

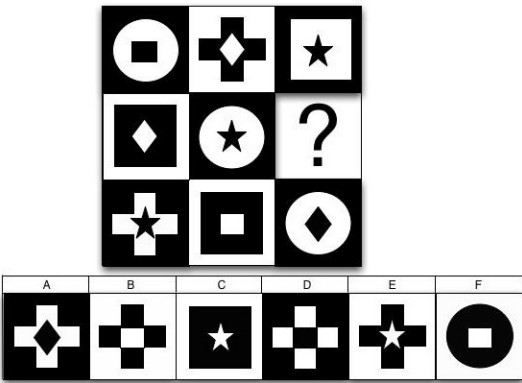

981

(Multiple choice, B)

982

983

984 *All the cubes below have a different image on each side. Select the choice that could repre-*
985 *sent a rotation of the cube labeled X*

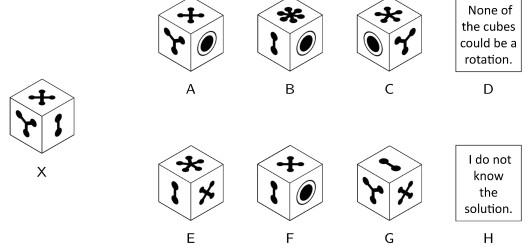

986

(Multiple choice, F)

987

988

989 *All the cubes below have a different image on each side. Select the choice that could repre-*
990 *sent a rotation of the cube labeled X*

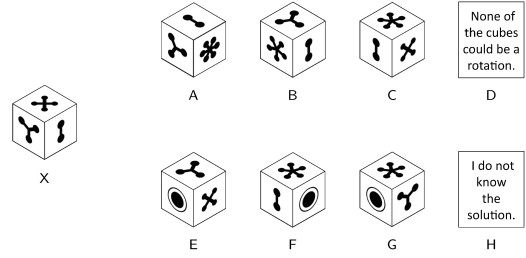

991

(Multiple choice, A)

992

993

994 *All the cubes below have a different image on each side. Select the choice that could repre-*

*sent a rotation of the cube labeled X*

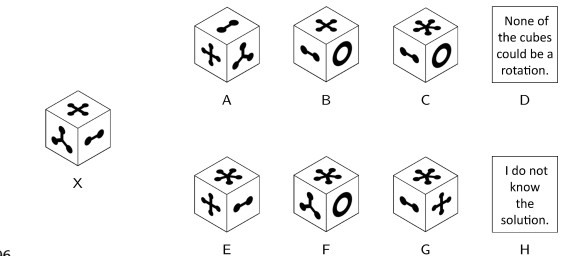

(Multiple choice, B)

Score: fluid intelligence (number of correct responses).

### A3.3.3 Crystallized intelligence test (Krefeld-Schwalb et al. 2024)

Questions (response options with correct answer in *italics*):
*Synonym: Select the word that is most nearly the same in meaning as CONCUR (*acquiesce*, extricate, divulge, concoct, ransack)
*Synonym: Select the word that is most nearly the same in meaning as CONFISCATE (harass, repulse, console, *appropriate*, congregate)
*Synonym: Select the word that is most nearly the same in meaning as SOLICIT (purge, spurn, entrance, exert, *beseech*)
*Synonym: Select the word that is most nearly the same in meaning as FURTIVE (ecstatic, heinous, *stealthy*, flimsy, facile)
*Synonym: Select the word that is most nearly the same in meaning as ASTUTE (bizarre, ascetic, *sagacious*, lineal,irritable)
*Synonym: Select the word that is most nearly the same in meaning as COVET (*crave*, claim, avenge, clutch, comply)
*Synonym: Select the word that is most nearly the same in meaning as OSCILLATE (premeditate, irradiate, *vacilate*,[8] recapitulate, furbish)
*Synonym: Select the word that is most nearly the same in meaning as INDOLENT (contrite, inexhaustible, impervious, arduous, *slothful*)
*Synonym: Select the word that is most nearly the same in meaning as DISPARITY (despondency, mediocrity, serenity, *incongruity*, assiduity)
*Synonym: Select the word that is most nearly the same in meaning as INDIGENT (refractory, fiscal, *destitute*, tolerable, diligent)
*Antonym: Select the word that is most nearly the opposite in meaning to SATED (*famished*, finished, finicky, fulfilled, fortunate)
*Antonym: Select the word that is most nearly the opposite in meaning to COMPLAISANT (distasteful, egoistical, alone, ugly, *recalcitrant*)
*Antonym: Select the word that is most nearly the opposite in meaning to ALOOF (happy, deadly, *gregarious*, manly, varied)
*Antonym: Select the word that is most nearly the opposite in meaning to ABOMINATE (*adore*, despair, abate, deplore, attach)
*Antonym: Select the word that is most nearly the opposite in meaning to VERBOSE (garrulous, magnificent, grandiloquent, *taciturn*, calculating)
*Antonym: Select the word that is most nearly the opposite in meaning to DEARTH (birth, brevity, *abundance*, splendor, renaissance)
*Antonym: Select the word that is most nearly the opposite in meaning to CORPULENT (sallow, affiliated, *emaciated*, entrepreneur, anemic)
*Antonym: Select the word that is most nearly the opposite in meaning to GERMANE (teutonic, healthful, *irrelevant*, massive, puny)
*Antonym: Select the word that is most nearly the opposite in meaning to VACUOUS (bankrupt, loose, livid, superficial, *profound*)
*Antonym: Select the word that is most nearly the opposite in meaning to SPORADIC (germinal,

---

[8]Word was misspelled unintentionally in survey.

antiseptic, *incessant*, summery, wintry)
Score: crystallized intelligence (number of correct responses)

### A3.3.4 Syllogisms test (*Markovits and Nantel 1989*)

*Suppose that it is true that:*
*All the XAR's are YOF's.*
*With this in mind, answer the following questions.*

Questions (response options with correct answer in *italics*):
*If a glock is a XAR, you can say:* (*that it is certain that the glock is a YOF*, that it is certain that the glock is not a YOF, that it is not certain whether the glock is a YOF or not)
*If a glock is not a XAR, you can say:* (that it is certain that the glock is a YOF, that it is certain that the glock is not a YOF, *that it is not certain whether the glock is a YOF or not*)
*If a koy is a YOF, you can say:* (that it is certain that the koy is a XAR, that it is certain that the koy is not a XAR, *that it is not certain whether the koy is a XAR or not*)
*If a koy is not a YOF, you can say:* (that it is certain that the koy is a XAR, *that it is certain that the koy is not a XAR*, that it is not certain whether the koy is a XAR or not)

*You are now going to receive a series of eight problems. You must decide whether the stated conclusion follows logically from the premises or not. You must suppose that the premises are all true and limit yourself only to the information contained in these premises.*

*Premise 1: All things that are smoked are good for the health. Premise 2: Cigarettes are smoked. Conclusion: Cigarettes are good for the health. Does the conclusion follow logically from the premises?* (*yes*, no)
*Premise 1: All animals love water. Premise 2: Plants do not love water. Conclusion: Plants are not animals. Does the conclusion follow logically from the premises?* (*yes*, no)
*Premise 1: All animals with four legs are dangerous. Premise 2: Poodles are not dangerous. Conclusion: Poodles do not have four legs. Does the conclusion follow logically from the premises?* (*yes*, no)
*Premise 1: All eastern countries are communist. Premise 2: China is not an eastern country. Conclusion: China is not communist. Does the conclusion follow logically from the premises?* (yes, *no*)
*Premise 1: All flowers have petals. Premise 2: Roses have petals. Conclusion: Roses are flowers. Does the conclusion follow logically from the premises?* (yes, *no*)
*Premise 1: All mammals swim. Premise 2: Whales are mammals. Conclusion: Whales swim. Does the conclusion follow logically from the premises?* (*yes*, no)
*Premise 1: All unemployed people are poor. Premise 2: Rockefeller is not unemployed. Conclusion: Rockefeller is not poor. Does the conclusion follow logically from the premises?* (yes, *no*)
*Premise I: All things that have a motor need oil. Premise 2: Bicycles need oil. Conclusion: Bicycles have motors. Does the conclusion follow logically from the premises?* (yes, *no*)

Score: syllogism (number of correct responses).

### A3.3.5 Overconfidence (*Dean and Ortoleva 2019*)

Question: *You just answered 42 questions that measured your performance on various cognitive tests. How many of these questions do you think you answered correctly? Enter a whole number between 0 and 42.*
Response format: integer between 0 and 42.
Score: overconfidence (belief of own performance on 42 cognitive test questions in wave 1 - actual performance).

### A3.3.6 Overplacement (*Dean and Ortoleva 2019*)

Question: *How many of these questions do you think you people from a representative sample of the US adult population would answer correctly, on average? Enter a whole number between 0 and 42.*
Response format: integer between 0 and 42.
Score: overplacement (belief of own performance on 42 cognitive test questions in wave 1 - belief of average performance).

### A3.3.7 Financial literacy test (*Johnson et al. 2019*)

Multiple-choice questions (response options with correct answer in *italics*):
*Do you think that the following statement is true or false? "A 15-year mortgage typically requires higher monthly payments than a 30-year mortgage, but the total interest paid over the life of the loan will be less."* (*True*, False)
*Imagine that the interest rate on your saving account was 1% per year and inflation was 2% per year. After 1 year, would you be able to buy:* (More than today with the money on this account, Exactly the same as today with the money in this account, *Less than today with the money in this account*, Do not know)
*Normally, which asset described below displays the highest fluctuations over time?* (Savings account, *Stocks*, Bonds, Do not know)
*Do you think that the following statement is true or false? "If you were to invest $1,000 in a stock mutual fund, it would be possible to have less than $1,000 when you withdraw your money."* (*True*, False)
*When an investor spreads their money among different assets, the risk of losing a lot of money:* (Increases, *Decreases*, Stays the same, Do not know)
*Considering a long time period (for example 10 or 20 years), which asset described below normally gives the highest return?* (Savings account, *Stocks*, Bonds, Do not know)
*Do you think that the following statement is true or false? "After age 70 and a half, you have to withdraw at least some money from your 401(k) plan or IRA."* (True, *False*)[9]
*Suppose you owe $3,000 on your credit card. You pay a minimum payment of $30 each month. At an Annual Percentage Rate of 12% (or 1% per month), how many years would it take to eliminate your credit card debt if you made no additional new charges? Enter a number of years, or "never" if the debt will never be eliminated.* (open-ended, *never*).
Score: financial literacy (number of correct responses)

### A3.3.8 Numeracy test (*Johnson et al. 2019*)

Questions (correct answer):
*Imagine that we roll a fair, six-sided die 1,000 times. Out of 1,000 rolls, how many times do you think the die would come up as an even number? Which is the most likely outcome? Enter a number from 0 to 1,000.* (500)
*In the BIG BUCKS LOTTERY, the chance of winning a $10.00 prize is 1%. What is your best guess about how many people would win a $10.00 prize if 1,000 people each buy a single ticket from BIG BUCKS? Enter a number from 0 to 1,000.* (10)
*If the chance of getting a disease is 20 out of 100, this would be the same as having a [blank]% chance of getting the disease. Enter a percentage from 0 to 100.* (20)
*In the ACME PUBLISHING SWEEPSTAKES, the chance of winning a car is 1 in 1,000. What percent of tickets of ACME PUBLISHING SWEEPSTAKES win a car? Enter a percentage from 0 to 100.* (0.1)
*If the chance of getting a disease is 10%, how many people would be expected to get the disease out of 1,000? Enter a number from 0 to 1,000.* (100)
*If it takes 5 machines 5 minutes to make 5 widgets, how long would it take 100 machines to make 100 widgets? Enter a number of minutes.* (5)
*A bat and ball cost $1.10 in total. The bat costs $1.00 more than the ball. How much does the ball*

---

[9]Legislation has evolved since the questionnaire was initially developed.

*cost? Enter a number in dollars.* (0.05)

*In a lake, there is a patch of lily pads. Every day, the patch doubles in size. If it takes 48 days for the patch to cover the entire lake, how long would it take for the patch to cover half of the lake? Enter a number of days.* (47)

Score: numeracy (number of correct responses)

### A3.3.9 Deductive certainty of Modus Ponens test (*Stanovich and West 2008*)

*You are now going to receive a series of four problems. You must decide whether the stated conclusion follows logically from the premises or not.*
*You must suppose that the premises are all true and limit yourself only to the information contained in these premises.*

Questions (correct answer):

*Premise 1: If there is a postal strike, then unemployment will double.*
*Premise 2: There is a postal strike.*
*Conclusion: Unemployment will double.*
*Does the conclusion follow logically from the premises?* (Yes)

*Premise 1: If the winter is harsh, then there will be a flu epidemic.*
*Premise 2: The winter is harsh.*
*Conclusion: There will be a flu epidemic.*
*Does the conclusion follow logically from the premises?* (Yes)

*Premise 1: If a car is a Honda, then it is expensive.*
*Premise 2: A car is a Honda.*
*Conclusion: The car is expensive.*
*Does the conclusion follow logically from the premises?* (Yes)

*Premise 1: If a person eats hamburgers, then they will get cancer.*
*Premise 2: A person eats hamburgers.*
*Conclusion: The person will get cancer.*
*Does the conclusion follow logically from the premises?* (Yes)

Score: deductive certainty score (number of correct responses)

### A3.3.10 Forward Flow (free associations) (*Gray et al. 2019*)

[Each participant randomly assigned to one of the following seed word: candle, snow, toaster, paper, table, bear]
*On this page, starting with the word {seed word}, your job is to write down the next word that follows in your mind from the previous word. Please put down only single words, and do not use proper nouns (such as names, brands, etc.). There is no right or wrong answer, just write the words as they come to your mind.*
Response format: 20 text boxes (first pre-populated with the seed word).
Score: forward flow (average pairwise cosine similarity based on the 20 Word2vec embeddings).

### A3.3.11 Wason Selection Task (*Klauer et al. 2007*)

*Imagine you see a number of cards from a set of cards. Each card in the set has a capital letter on one side and a number on the other. Naturally, only one side is visible in each case.*
*For the set of cards, a rule has been stated: If there is an A on the letter side of the card, then there is a 3 on the number side.*
*Four cards were drawn. Below is the information visible for each card (letter or number).*
*Which of the following card(s) would have to be turned over in order to test the truth or falsity of the*

1198 *rule?*
1199 Options (correct/incorrect): A (correct); F (incorrect); 3 (incorrect); 7 (correct)
1200 Score: Wason Selection Task score (number of correct responses)

1201 **A3.4   Economic preferences**

1202 *A3.4.1   Ultimatum Game (Güth et al. 1982)*

1203
1204 Send question (options): *Suppose you were given $5 and had to offer to another (anonymous)*
1205 *person a way to split the money. You would propose how much of this money to keep for yourself and*
1206 *how much to send them.*
1207 *Then, the other person would have to decide whether or not to accept your offer. If they accept your*
1208 *offer, you would each receive the amount specified in your offer.*
1209 *If they reject your offer, you would both receive nothing.*
1210 *In this scenario, how much would propose to keep for yourself and how much would you propose to*
1211 *send to the other person?* ($0 for myself, $5 to the other person; $1 for myself, $4 to the other
1212 person.; $2 for myself, $3 to the other person; $3 for myself, $2 to the other person; $4 for
1213 myself, $1 to the other person; $5 for myself, $0 to the other person.
1214 Measure: ultimatum-send (% of total amount sent).

1215
1216 *Suppose now that you are playing this game as the other person, i.e., the receiver.*

1217
1218 *For each offer made by the sender, would you accept or reject the offer?*
1219 Receive questions (options): *If the person offers to keep $0 for themselves and send me $5:*
1220 (I would accept the offer: $0 for other, $5 for me; I would reject the offer: $0 for both)
1221 *If the person offers to keep $1 for themselves and send me $4:*
1222 (I would accept the offer: $1 for other, $4 for me; I would reject the offer: $0 for both)
1223 *If the person offers to keep $2 for themselves and send me $3:*
1224 (I would accept the offer: $2 for other, $3 for me; I would reject the offer: $0 for both)
1225 *If the person offers to keep $3 for themselves and send me $2:*
1226 (I would accept the offer: $3 for other, $2 for me; I would reject the offer: $0 for both)
1227 *If the person offers to keep $4 for themselves and send me $1:*
1228 (I would accept the offer: $4 for other, $1 for me; I would reject the offer: $0 for both)
1229 *If the person offers to keep $5 for themselves and send me $0:*
1230 (I would accept the offer: $5 for other, $0 for me; I would reject the offer: $0 for both)
1231
1232 Measure: ultimatum-receiver (acceptance probability).

1233 *A3.4.2   Mental accounting (Thaler 1985)*

1234
1235 Questions (options):
1236 *Person A was given tickets to lotteries involving the World Series. They won $50 in one lottery and*
1237 *$25 in the other.*
1238 *Person B was given a ticket to a single, larger World Series lottery. They won $75.*
1239 *Who was happier?* (Person A, Person B)

1240
1241 *Person A received a letter from the IRS saying that they made a minor arithmetical mis-*
1242 *take on their tax return and owed $100. They received a similar letter the same day from their state*
1243 *income tax authority saying they owed $50. There were no other repercussions from either mistake.*
1244 *Person B received a letter from the IRS saying that they made a minor arithmetical mistake on their*
1245 *tax return and owed $150. There were no other repercussions from either mistake.*
1246 *Who was more upset?* (Person A, Person B)

1247
1248 *Person A bought their first New York State lottery ticket and won $100.  Also, in a freak*
1249 *accident, they damaged the rug in their apartment and had to pay the landlord $80.*
1250 *Person B bought their first New York State lottery ticket and won $20. Who was happier?* (Person
1251 A, Person B)

*Person A's car was damaged in a parking lot. They had to spend $200 to repair the damage. The same day the car was damaged, they won $25 in the office football pool.*
*Person B's car was damaged in a parking lot. They had to spend $175 to repair the damage.*
*Who was more upset?* (Person A, Person B)

Measure: mental accounting score (percentage of responses consistent with mental accounting predictions: A, A, B, B).

### A3.4.3   Discount (Dean and Ortoleva 2019)

*Please choose between the following options. For each line in the list, you must choose between the option on the left and the option on the right. Note that on each line, the option on the left stays the same while the option on the right gets better as one goes down the list. You can select the option you would prefer receiving by clicking on the button next to that option.*

Choice 1: multiple price list. Left option: $6.00 in 6 weeks. Right option: $x in 5 weeks, with $x \in \{3.00, 4.00, 4.50, 5.00, 5.25, 5.50, 5.75, 6.00, 7.00\}$.
Measure 1: based on lowest value $x$ for which option on the right is preferred, compute equivalent annualized discount rate as $(\frac{6}{x})^{52/1} - 1$. (N\A if no switching).

Choice 2: multiple price list. Left option: $8.00 in 7 weeks. Right option: $x in 6 weeks, with $x \in \{4.00, 5.00, 6.00, 7.00, 7.25, 7.50, 7.75, 8.00, 9.00\}$.
Measure 2: equivalent annualized discount rate: $(\frac{8}{x})^{52/1} - 1$. (N\A if no switching).

Choice 3: multiple price list. Left option: $10.00 in 7 weeks. Right option: $x in 5 weeks, with $x \in \{5.00, 6.00, 7.00, 8.00, 8.50, 9.00, 9.25.9.50, 9.75, 10.00, 11.00\}$.
Measure 3: equivalent annualized discount rate: $(\frac{10}{x})^{52/2} - 1$. (N\A if no switching).

Measure: discount rate (average of three measures).

### A3.4.4   Present bias (Dean and Ortoleva 2019)

Three Present Discount questions:
*Please choose between the following options. For each line in the list, you must choose between the option on the left and the option on the right. Note that on each line, the option on the left stays the same while the option on the right gets better as one goes down the list. You can select the option you would prefer receiving by clicking on the button next to that option.*

Choice 1: multiple price list. Left option: $6.00 in 1 weeks. Right option: $x now, with $x \in \{3.00, 4.00, 4.50, 5.00, 5.25, 5.50, 5.75, 6.00, 7.00\}$.
Measure 1: based on lowest value $x$ for which option on the right is preferred, compute equivalent annualized discount rate as $(\frac{6}{x})^{52/1} - 1$. (N\A if no switching).

Choice 2: multiple price list. Left option: $8.00 in 1 weeks. Right option: $x now, with $x \in \{4.00, 5.00, 6.00, 7.00, 7.25, 7.50, 7.75, 8.00, 9.00\}$.
Measure 2: equivalent annualized discount rate: $(\frac{8}{x})^{52/1} - 1$. (N\A if no switching).

Choice 3: multiple price list. Left option: $10.00 in 2 weeks. Right option: $x now, with $x \in \{5.00, 6.00, 7.00, 8.00, 8.50, 9.00, 9.25.9.50, 9.75, 10.00, 11.00\}$.
Measure 3: equivalent annualized discount rate: $(\frac{10}{x})^{52/2} - 1$. (N\A if no switching).

For each of the three Discount questions, consider the corresponding Present Discount question. Let $x$ (respectively, $z$) be the lowest value for which option on the right is preferred in the Discount (respectively, Present Discount) question, and let $y$ be the

larger-later amount. Present bias for that pair of question is computed as $\frac{x-z}{y}$. Present bias is measured as the average across the three pairs of questions.

### A3.4.5   Risk aversion ([Dean and Ortoleva 2019](#))

Question 1: *In this question, the LOTTERY is a 50% chance of winning $6 and a 50% chance of winning $0. A graphical representation of the lottery is found below.*
*Suppose you are given the option to exchange this lottery for certain amounts of money.*
*Please choose between the following options. For each line in the list, you must choose between the option on the left and the option on the right. Note that on each line, the option on the left stays the same while the option on the right gets better as one goes down the list.*

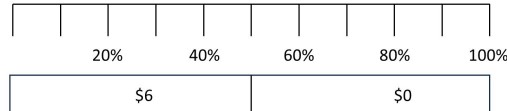

(When creating digital twins this image was replaced with the following text: *The image displays a probability scale from 0% to 100%, marked at intervals of 20%. Below the scale are two side-by-side boxes: one labeled "$6" on the left that extends from 0% to 50%, and the other labeled* "$3" *on the right, that extends from 50% to 100%.* Other lottery illustrations were described similarly.)
*You can select the option you would prefer receiving by clicking on the button next to that option.*
Choice 1: multiple price list. Left option: "Lottery." Right option: $x$ where
$x \in \{0.50, 1.00, 1.25, 1.50, 1.75, 2.00, 2.25, 2.50, 2.75, 3.00, 3.25, 3.50, 4.00, 5.00\}$
Measure 1: $\frac{EV-CE}{EV}$ where $EV$ is the lottery's expected value and $CE$ is the lowest amount for which the option on the right is chosen. (N\A if no switching).

Question 2: same as Question 1, with LOTTERY a 50% chance of winning $8 and a 50% chance of winning $2.
Choice 2: Same as Choice 1, where
$x \in \{2.50, 3.00, 3.25, 3.50, 3.75, 4.00, 4.25, 4.50, 4.75, 5.00, 5.25, 5.50, 6.00, 7.00\}$
Measure 2: Same formula as Question 1.

Question 3: same as Question 1, with LOTTERY a 50% chance of winning $10 and a 50% chance of winning $0.
Choice 3: Same as Choice 2
Measure 3: Same formula as Question 1.

Measure: risk aversion (average from the three measures).

### A3.4.6   Loss aversion ([Dean and Ortoleva 2019](#))

Three questions mirroring the Risk Aversion questions in the loss domain (e.g., first question is: *In this question, the LOTTERY is a 50% chance of LOSING $6 (as indicated by the minus sign before $6) and a 50% chance of winning $0. A graphical representation of the lottery is found below. Suppose you are given the option to exchange this lottery for certain amounts of money. The alternative also involves losing money, as indicated by the minus sign. Please choose between the following options. For each line in the list, you must choose between the option on the left and the option on the right. Note that on each line, the option on the left stays the same while the option on the right gets better as one goes down the list. You can select the option you would prefer receiving by clicking on the button next to that option.*)

Estimate a constant relative risk aversion (CRRA) coefficient in gain domain based each of the three gain questions.

Estimate a constant relative risk aversion (CRRA) coefficient function in loss domain based on each of the three loss questions.

Elicit value $x$ that makes participant indifferent between $0 for sure and a 50/50 gamble between $x and -$8: *In this question, the LOTTERY is a 50% chance of LOSING $8 (as indicated by the minus sign before $8) and a 50% chance of winning a value x. A graphical representation of the lottery is found below. Suppose you are given the option to exchange this lottery for the certainty of winning $0. Please choose between the following options. For each line in the list, you must choose between the option on the left and the option on the right. Note that on each line, the option on the left gets better as one goes down the list while the option on the right stays the same. You can select the option you would prefer receiving by clicking on the button next to that option.* Choice: multiple price list. Left option: Lottery with $x \in \{7.00, 8.00, 9.00, 10.00, 11.00, 12.00, 13.00, 14.00, 15.00, 16.00, 17.00, 18.00, 19.00, 20.00\}$. Right option: $0.

Measure: Loss aversion $\lambda$ estimated as the additional slope of the utility function in the loss domain relative to the gain domain that is necessary to match this last choice, conditional on the slopes estimated separately in the two domains. That is, we calculate: $\lambda = \frac{-U_G(x)}{U_L(-8)}$ (where $x$ is the lowest amount for which the option on the right is chosen) for each of the three pairs of questions, and take the average.

### A3.4.7   Trust game ([Dean and Ortoleva 2019](#))

Questions (response options):
*Suppose you were given $5 and had to decide how much of this money to keep for yourself and how much to send to another (anonymous) person.*
*Any amount you send to the other person would then be tripled. That is, if you send $1, this becomes $3. If you send $2, this becomes $6, etc.*
*Then, the other person would have to decide how much of that money to keep and how much to return to you. That is, if you send $1, this would become $3 and the other person would have to decide how much of this $3 to keep for themself and how much to send back to you.*
*In this scenario, how much would keep for yourself and how much would you send to the other person?*
("I would keep $0 for myself and send $5 to the other person" to $5 for myself and $0 to other, in $1 increments.)
Measure: trust-send (percentage of total amount sent)

*Suppose now that you are playing this game as the other person, i.e., the receiver.*
*For each amount that you may receive, how much would you keep for yourself and how much would send back to the other person?*
*If the person sends me $5 (which would become $15):*
("I would keep $0 for myself and send $15 to the other person" to keep $15 and send $0, in $1 increments)
*If the person sends me $4 (which would become $12):*
("I would keep $0 for myself and send $12 to the other person" to keep $12 and send $0, in $1 increments)
*If the person sends me $3 (which would become $9):*
("I would keep $0 for myself and send $9 to the other person" to keep $9 and send $0, in $1 increments)
*If the person sends me $2 (which would become $6):*
("I would keep $0 for myself and send $6 to the other person" to keep $6 and send $0, in $1 increments)
*If the person sends me $1 (which would become $3):*
("I would keep $0 for myself and send $3 to the other person" to keep $3 and send $0, in $1 increments)

Measure: trust-receiver (average percentage returned)

1410

Thought listing - sender: *We are now interested in what you were thinking about while deciding how much of the money to keep for yourself and how much to send to another (anonymous) person, and when deciding how much would you keep for yourself and how much would send back to the other person.*

*Any thought is fine; simply list what it was that you were thinking about while answering the questions.*

*Below, please write down the first thought that you had in the first box, the second thought you had in the second box, etc.*

*Please write only one idea per box. You should try to write only the thoughts that you were thinking during the task.*

*Please state your thoughts concisely...one phrase is sufficient. Ignore spelling, grammar, and punctuation.*

*Please be completely honest and list all of thoughts that you had.*

*Don't worry if you don't fill every space. Just write down whatever thoughts you had while making the decision.*

Response format: 6 text boxes (responses optional)

Thought listing - receiver: *Second, please list the thoughts you had when deciding how much would you keep for yourself and how much would send back to the other person:*

Response format: 6 text boxes (responses optional)

### A3.4.8 Dictator Game (Baron and Hershey 1988)

Question (options): *Suppose you were given $5 and had to split the money between yourself and another (anonymous) person. You and you only would decide how to split the money, the other person would need to accept your offer.*

*In this scenario, how much would keep for yourself and how much would you send to the other person?*

($0 for myself, $5 to the other person.; $1 for myself, $4 to the other person.; $2 for myself, $3 to the other person.; $4 for myself, $1 to the other person.; $5 for myself, $0 to the other person.)

Measure: Dictator-send (percentage of total amount sent).

Thought-listing:

*We are now interested in what you were thinking about while deciding how much of the money to keep for yourself and how much to send to another (anonymous) person.*

*Any thought is fine; simply list what it was that you were thinking about while answering the questions.*

*Below, please write down the first thought that you had in the first box, the second thought you had in the second box, etc. Please write only one idea per box. You should try to write only the thoughts that you were thinking during the task.*

*Please state your thoughts concisely...one phrase is sufficient. Ignore spelling, grammar, and punctuation.*

*Please be completely honest and list all of thoughts that you had. Don't worry if you don't fill every space. Just write down whatever thoughts you had while making the decision.*

*Please list the thoughts you had when deciding how much of the money to keep for yourself and how much to send to another (anonymous) person:*

Response format: 6 text boxes (responses optional)

## A3.5 Heuristics and biases - between subject

### A3.5.1 Base rate problem (Kahneman and Tversky 1973)

Conditions: 30 Engineers, 70 Engineers

Question: *A panel of psychologist have interviewed and administered personality tests to [30 engineers and 70 lawyers, 70 engineers and 30 lawyers], all successful in their respective fields.*

On the basis of this information, thumbnail descriptions of the [30 engineers and 70 lawyers, 70 engineers and 30 lawyers] have been written. Below is one description, chosen at random from the 100 available descriptions.

*Jack is a 45-year-old man. He is married and has four children. He is generally conservative, careful, and ambitious. He shows no interest in political and social issues and spends most of his free time on his many hobbies which include home carpentry, sailing, and mathematical puzzles.*

*The probability that Jack is one of the [30, 70] engineers in the sample of 100 is _%. Please indicate the probability on a scale from 0 to 100.*

Response scale: slider (0-100).

Results:

Wave 1: Unlike Tversky and Kahneman (1974) who find no difference between conditions, we find that the average probability judgment was significantly lower in the "30 Engineers" condition compared to the "70 Engineers" condition ($Prob_{30} = 52.17\%$, $Prob_{70} = 68.01\%$, $t = -15.78$, $p < 0.01$).

Wave 4: similar results ($Prob_{30} = 52.39\%$, $Prob_{70} = 70.71\%$, $t = -19.36$, $p < 0.01$).

### A3.5.2 Outcome bias (Baron and Hershey 1988)

Conditions: success, failure

Question: *A 55-year-old man had a heart condition. He had to stop working because of chest pain. He enjoyed his work and did not want to stop. His pain also interfered with other things, such as travel and recreation. A type of bypass operation would relieve his pain and increase his life expectancy from age 65 to age 70. However, [8% of the people who have this operation die from the operation itself, 2% of the people who have this operation die from the operation itself]. His physician decided to go ahead with the operation.*

*The operation [succeeded, did not succeed and the patient died].*

*Evaluate the physician's decision to go ahead with the operation.*

Response scale: Incorrect, a very bad decision (-3); Incorrect, all things considered (-2); Incorrect, but not unreasonable (-1); The decision and its opposite are equally good (0); Correct, but the opposite would be reasonable tocfo (1); Correct, all things considered (2); Clearly correct, an excellent decision (3).

Results:

Wave 1: Similar to Baron and Hershey (1988), we find that the average evaluation is more favorable in the "success" condition compared to the "failure" condition ($M_{success} = 1.66$, $M_{failure} = 0.88$, $t = 13.55$, $p < 0.001$).

Wave 4: similar results ($M_{success} = 1.64$, $M_{failure} = 1.04$, $t = 11.15$, $p < 0.001$).

### A3.5.3 Sunk cost fallacy (Stanovich and West 2008)

Conditions: no, yes

Question - no sunk cost condition: *Imagine that Coffee Connection sells coffee for $1.50 per cup. Java Coffee, a competitor, sells coffee for just $2.00 per cup.*

*Although the Coffee Connection store is ten minutes away by car, Java Coffee is only about 1/2 block from your apartment.*

*Assuming that you only buy coffee from these two places and that you like the coffee sold in both places the same, how many of your next 20 coffee purchases would be from Java Coffee?*

*Enter a number between 0 and 20.*

Question - sunk cost condition: *Imagine that you just paid $50 for a Coffee Connection discount card that allows you to buy coffee for 50% off the regular price of $3.00 (i.e., you pay $1.50).*

*Soon after you purchased the Coffee Connection discount card, Java Coffee, a competitor, opened a new store that sells coffee for just $2.00 per cup.*

*Although the Coffee Connection store is ten minutes away by car, Java Coffee is only about 1/2 block from your apartment.*

*Assuming that you only buy coffee from these two places and that you like the coffee sold in both places the same, how many of your next 20 coffee purchases would be from Java Coffee?*

*Enter a number between 0 and 20.*

Response format: integer between 0 and 20.

Results:

Wave 1: Similar to Stanovich and West (2008), we find that the average number of purchases is lower in the "sunk cost" condition compared to the "no sunk cost" condition ($M_{sunk\ cost} = 10.64$, $M_{no\ sunk\ cost} = 14.88$, $t = 16.20$, $p < 0.001$).

Wave 4: similar results ($M_{sunk\ cost} = 11.01$, $M_{no\ sunk\ cost} = 14.46$, $t = 14.07$, $p < 0.001$).

### A3.5.4 Allais problem (*Stanovich and West 2008*)

Conditions: Form 1, Form2

Choice between two gambles:

Form 1:

One million dollars for sure (A)

89% probability of one million dollars, 10% probablity of five million dollars, 1% probability of nothing (B)

Form 2:

11% probability of one million dollars, 89% probability of nothing (C)

10% probability of five million dollars, 90% probability; of nothing (D)

Results:

Wave 1: Similar to Stanovich and West (2008), we find that a significant majority of participants chose Option A in Form 1 ($Prob(A) = 69.2\%$, $p < 0.001$), and a significant majority chose Option D in Form 2 ($Prob(D) = 57.2\%$, $p < 0.001$).

Wave 4: similar results ($Prob(A) = 63.6\%$, $p < 0.001$, $Prob(D) = 62.3\%$, $p < 0.001$).

### A3.5.5 Framing problem (*Tversky and Kahneman 1981*)

Conditions: gain framing, loss framing

Question: *Imagine that the U.S. is preparing for the outbreak of an unusual disease, which is expected to kill 600 people.*

*Two alternative programs to combat the disease have been proposed. Assume that the exact scientific estimate of the consequences of the programs are as follows: If Program A is adopted, [200 people will be saved, 400 people will die].*

*If Program B is adopted, there is 1/3 probability that [600 people will be saved, nobody will die] and 2/3 probability that [no people will be saved, 600 people will die].*

*Which of the two programs would you favor?*

Response scale: I strongly favor program A (1), I favor program A (2), I slightly favor program A (3), I slightly favor program B (4), I favor program B (5), I strongly favor program B (6)

Results:

Wave 1: Similar to Tversky and Kahneman (1981), we find that the loss frame resulted in a greater preference for the risky option B ($M_{gain} = 2.85$, $M_{loss} = 3.84$, $t = -17.35$, $p < 0.001$).

Wave 4: Similar results ($M_{gain} = 2.83$, $M_{loss} = 3.76$, $t = -17.25$, $p < 0.001$).

### A3.5.6 Conjunction problem (Linda) (*Tversky and Kahneman 1983*)

Conditions: bank teller, feminist bank teller

*Linda is 31 years old, single, outspoken, and very bright. She majored in philosophy. As a student, she was deeply concerned with issues of discrimination and social justice, and also participated in anti-nuclear demonstrations.*

*Please complete the statements below.*

Response scale: Extremely improbable (1), Very improbable (2), Somewhat probable (3), Moderately probable (4), Very probable (5), Extremely probable (6)

Items: *It is __ that Linda is a teacher in an elementary school; It is __ that Linda works in a bookstore*

*and takes Yoga classes; It is __ that Linda is [a bank teller, a bank teller and is active in the feminist movement]*

Results:

Wave 1: consistent with Tversky and Kahneman (1983), we find that Linda was judged more probably a feminist bank teller than a bank teller ($M_{bank\ teller} = 2.43$, $M_{feminist\ bank\ teller} = 3.38$, $t = -18.83$, $p < 0.001$)

Wave 2: similar results ($M_{bank\ teller} = 2.52$, $M_{feminist\ bank\ teller} = 3.35$, $t = -17.82$, $p < 0.001$)

### A3.5.7  Anchoring and adjustment (*Tversky and Kahneman 1974, Epley et al. 2004*)

Conditions: large anchor, small anchor

Question 1: *Do you think there are more or fewer than [65,12] African countries in the United Nations?* (more, fewer)

Question 2: *How many African countries do you think are in the United Nations?* (numerical answer)

Results:

Wave 1: consistent with Tversky and Kahneman (1974), we find that the larger anchor resulted in higher estimates of the number of African countries in the United Nations ($M_{large\ anchor} = 48.22$, $M_{small\ anchor} = 26.36$, $t = 4.57$, $p < 0.001$).

Wave 1: similar results ($M_{large\ anchor} = 50.82$, $M_{small\ anchor} = 32.02$, $t = 13.57$, $p < 0.001$).

Question 1: *Is the tallest redwood tree in the world more or less than [1000,85] feet tall?* (more, less)

Question 2: *How tall do you think the tallest redwood tree in the world is? Enter a number of feet.* (numerical answer)

Results:

Wave 1: consistent with Tversky and Kahneman (1974), we find that the larger anchor resulted in higher estimates of the height of the tallest redwood tree in the world ($M_{large\ anchor} = 839.18$, $M_{small\ anchor} = 165.00$, $t = 22.03$, $p < 0.001$).

Wave 2: similar results ($M_{large\ anchor} = 824.01$, $M_{small\ anchor} = 213.17$, $t = 20.80$, $p < 0.001$).

### A3.5.8  Absolute vs. relative savings (*Stanovich and West 2008*)

Conditions: large percentage (calculator), small percentage (jacket)

Question: *Imagine that you go to purchase a [calculator for \$30, jacket for \$250].*

*The [calculator,jacket] salesperson informs you that the [calculator,jacket] you wish to buy is on sale for [\$20,\$240] at the other branch of the store which is ten minutes away by car.*

*Would you drive to the other store?* (Yes, No)

Results:

Wave 1: consistent with Stanovich and West (2008), we find that more participants were willing to make the trip to save \$10 for the calculator (large percentage) than for the jacket (small percentage) ($Prop_{large\ percentage} = 0.74$, $Prop_{small\ percentage} = 0.34$, $\chi^2 = 319.10$, $p < 0.001$).

Wave 4: similar results ($Prop_{large\ percentage} = 0.73$, $Prop_{small\ percentage} = 0.29$, $\chi^2 = 388.43$, $p < 0.001$).

### A3.5.9  Myside bias (*Stanovich and West 2008*)

Conditions: German car, Ford Explorer

Question: *According to a comprehensive study by the U.S. Department of Transportation, [ a*

 *particular German car is, Ford Explorers are] 8 times more likely than a typical family car to kill*
*occupants of another car in a crash.*
*The [U.S. Department of Transportation, Department of Transportation in Germany] is considering*
*recommending a ban on the sale of [this German car, the Ford Explorer in Germany].*
*Do you think that [the U.S., Germany] should ban the sale of the [German car, Ford Explorer]?*
Response scale: definitely no (1), no (2), probably no (3), probably yes (4), yes (5), definitely
yes (6)

Results:
Wave 1: consistent with Stanovich and West (2008), we find that participants were more
likely to think that the German car should be banned in the U.S. than they were to think that
the Ford Explorer should be banned in Germany ($M_{German\ car} = 4.46$, $M_{Ford\ Explorer} = 4.11$,
$t = 5.86$, $p < 0.001$).
Wave 4: similar results ($M_{German\ car} = 4.54$, $M_{Ford\ Explorer} = 4.10$, $t = 7.81$, $p < 0.001$).

### A3.5.10  Less is More (*Stanovich and West 2008*)

Conditions: Form A, Form B, Form C.
Question 1: *Please rate your level of disagreement or agreement with the following statement:*
*"I would find a game that had a 7/36 chance of winning $9 and a 29/36 chance of [winning nothing,*
*losing $0.05, losing $0.25] extremely attractive."*
Question 2: *Imagine that highway safety experts have determined that a substantial number of*
*people are at risk of dying in a type of automobile fire. A requirement that every car have a built-in*
*fire extinguisher (estimated cost, $300) would save [the 150 people, 98% of the 150 people, 95% of*
*the 150 people] who would otherwise die every year in this type of automobile fire.*
*Rate the following statement about yourself: I would be supportive of this requirement.*
Question 3: *You have recently graduated from university, obtained a good job, and are buying*
*a new car. A newly designed seatbelt has just become available that would save the lives of [the*
*500 drivers, 98% of the 500 drivers, 95% of the 500 drivers] a year who are involved in a type of*
*head-on-collision. (Approximately half of these fatalities involve drivers who were not at fault.) The*
*newly designed seatbelt is not yet standard on most car models. However, it is available as a $500*
*option for the car model that you are ordering.*
*How likely is it that you would order your new car with this optional seatbelt?"*
Response scale (common for all three questions): Disagree strongly (1), Disagree a little (2),
Neither agree nor disagree (3), Agree a little (4), Agree strongly (5)

Results:
Wave 1: Consistent with Stanovich and West (2008), in each question we find that the
option with no possibility of loss (Form A) was rated as less appealing than either of
the options that contained the possibility of a loss ($M_A^1 = 2.06$, $M_B^1 = 2.89$, $M_C^1 = 2.86$,
$F(2, 2055) = 87.70$, $p < 0.001$; $M_A^2 = 4.03$, $M_B^2 = 4.29$, $M_C^2 = 4.30$, $F(2, 2055) = 13.75$,
$p < 0.001$; $M_A^3 = 4.44$, $M_B^2 = 4.75$, $M_C^3 = 4.74$, $F(2, 2055) = 11.19$, $p < 0.001$).
Wave 4: similar results ($M_A^1 = 2.15$, $M_B^1 = 3.15$, $M_C^1 = 3.01$, $F(2, 2055) = 112.01$, $p < 0.001$;
$M_A^2 = 3.97$, $M_B^2 = 4.22$, $M_C^2 = 4.27$, $F(2, 2055) = 14.63$, $p < 0.001$; $M_A^3 = 4.44$, $M_B^3 = 4.76$,
$M_C^3 = 4.79$, $F(2, 2055) = 14.41$, $p < 0.001$).

### A3.5.11  WTA/WTP – Thaler problem (*Stanovich and West 2008*)

Conditions: WTP-certainty, WTA-certainty, WTP-noncertainty

Question:
WTP-certainty: *Imagine that when you went to the movies last week, you were inadvertently*
*exposed to a rare and fatal virus.*
*The possibility of actually contracting the disease is 1 in 1,000, but once you have the illness there is*
*no known cure.*
*On the other hand, you can, readily and now, be given an injection that stops the development of the*

*illness.*

*Unfortunately, these injections are only available in very small quantities and are sold to the highest bidder.*

*What is the highest price you would be prepared to pay for such an injection? [You can get a long-term, low-interest loan if needed.]:*

WTA-certainty: *Imagine that a group of research scientists in the School of Medicine are running a laboratory experiment on a vaccine for a rare and fatal virus.*
*The possibility of actually contracting the disease from the vaccine is 1 in 1,000, but once you have the disease there is no known cure.*
*The scientists are seeking volunteers to test the vaccine on.*
*What is the lowest amount that you would have to be paid before you would take part in this experiment?*

WTP-noncertainty: *Imagine that when you went to the movies last week, you were inadvertently exposed to a rare and fatal virus.*
*The possibility of actually contracting the disease is 4 in 1,000, but once you have the illness there is no known cure.*
*On the other hand, you can, readily and now, be given an injection that reduces the possibility of contracting the disease to 3 in 1,000.*
*Unfortunately, these injections are only available in very small quantities and are sold to the highest bidder.*
*What is the highest price you would be prepared to pay for such an injection?*

Response scale: $10 (1), $100 (2), $1,000 (3), $10,000 (4), $50,000 (5), $100,000 (6), $250,000 (7), $500,000 (8), $1,000,000 (9), $5,000,000 or more (10)

Results:
Wave 1: consistent with Stanovich and West (2008), we find that the mean score in the WTA-certainty condition was significantly higher than the mean score in the WTP-certainty condition ($M_{WTA-certainty} = 6.82$, $M_{WTP-certainty} = 3.27$, $t = 26.25$, $p < 0.001$), and that the the mean score in the WTP-certainty condition was significantly higher than the mean score in the WTP-noncertainty condition ($M_{WTP-certainty} = 3.27$, $M_{WTP-noncertainty} = 2.20$, $t = 11.36$, $p < 0.001$).
Wave 4: similar results ($M_{WTA-certainty} = 7.24$, $M_{WTP-certainty} = 3.23$, $t = 31.15$, $p < 0.001$; $M_{WTP-certainty} = 3.23$, $M_{WTP-noncertainty} = 2.20$, $t = 11.22$, $p < 0.001$).

## A3.6    Heuristics and biases - within Subject

### A3.6.1    *False consensus (Furnas and LaPira 2024)*

Self question (asked just before economic preference questions in Wave 1, first question in Wave 4): *Would you support or oppose...*
Response scale: Strongly oppose, Somewhat oppose, Neither oppose nor support, Somewhat support, Strongly support
Items: *Placing a tax on carbon emissions?; Ensuring 40% of all new clean energy infrastructure development spending goes to low-income communities?; Federal investments to ensure a carbon-pollution free electricity sector by 2035?' A 'Medicare for All' system in which all Americans would get healthcare from a government-run plan?; A 'public option', which would allow Americans to buy into a government-run healthcare plan if they choose to do so?; Immigration reforms that would provide a path to U.S. citizenship for undocumented immigrants currently in the United States?; A law that requires companies to provide paid family leave for parents?; A 2% tax on the assets of individuals with a net worth of more than $50 million?; Increasing deportations for those in the US illegally?; Offering seniors healthcare vouchers to purchase private healthcare plans in place of traditional medicare coverage?*

Public question (last question in Wave 1, last question before pricing study in Wave 4): *What*

percentage of the public do you think supports the following policies? For each policy, choose a
number from 0% to 100%.
Response scale: slider (0-100)
Items: same as self question

Analysis: we run a two-way fixed effect regression
$Y_{ip} = \beta_1 StrongOpp_{ip} + \beta_2 SomewhatOpp_{ip} + \beta_3 SomewhatSupp_{ip} + \beta_4 StrongSupp_{ip} + \alpha_i + \gamma_p + \epsilon_{ip}$, where $Y_ip$ is respondent $i$'s misperception of public support for policy $p$
(predicted-actual public support, with actual support being the proportion of participants
who somewhat or strongly support the policy), $StrongOpp_{ip}$ etc. are dummy variables
indicating $i$'s support for $p$ (with "neither oppose nor support" as the reference), $\alpha_i$ is a
participant fixed effect, and $\gamma_p$ is a policy fixed effect.

Results:
Wave 1: consistent with Furnas and LaPira (2024), we find that the more participants support
a policy, the more they believe others support it ($\beta_1 = -13.07$, 95% CI=$[-14.15, -11.99]$;
$\beta_2 = -6.38$, 95% CI=$[-7.43, -5.36]$; $\beta_3 = 8.04$, 95% CI=$[7.25, 8.82]$; $\beta_4 = 16.65$, 95%
CI=$[15.84, 17.46]$).
Wave 4: similar results ($\beta_1 = -13.62$, 95% CI=$[-14.69, -12.55]$; $\beta_2 = -5.69$, 95%
CI=$[-6.68, -4.71]$; $\beta_3 = 9.53$, 95% CI=$[8.79, 10.28]$; $\beta_4 = 18.36$, 95% CI=$[17.58, 19.14]$).

### A3.6.2 Nonseparability of risk and benefits judgments (Stanovich and West 2008)

Benefits: *Please rate the following technology or products from "not at all beneficial" to "extremely
beneficial"*
Response scale: not at all beneficial (1), low benefit (2), slightly beneficial (3), neutral (4),
moderately beneficial (5), very beneficial (6), extremely beneficial (7)
Items: *bicycles, alcoholic beverages, chemical plants, pesticides*

Risks: *Please rate the following technology or products from "not at all risky" to "extremely risky"*
Response scale: not at all risky (1), low risk (2), slightly risky (3), neutral (4), moderately
risky (5), very risky (6), extremely risky (7)
Items: same as benefits question

Results:
Wave 1: we compute the correlation between benefit for each of the four items. Consistent
with Stanovich and West (2008), we find significant negative correlations for alcoholic
beverages ($r = -0.33$, $t = -15.97$, $p < 0.001$), chemical plants ($r = -0.29$, $t = -13.76$,
$p < 0.001$) and pesticides ($r = -0.37$, $t = -18.21$, $p < 0.001$). However, the correlation for
bicycles was close to 0 ($r = 0.00$, $t = 0.001$, $p = 1$). Note that Stanovich and West (2008) find
that the correlation for bicycles is significant only in their High-SAT group.
Wave 4: similar results ($r_{alcohol} = -0.36$, $t = -17.31$, $p < 0.001$; $r_{chemical} = -0.31$,
$t = -14.57$, $p < 0.001$; $r_{pesticides} = -0.37$, $t = -18.26$, $p < 0.001$; $r_{bycicle} = -0.01$, $t = -0.22$,
$p = 0.83$).

### A3.6.3 Omission bias (Stanovich and West 2008)

Question: *Imagine that there will be a deadly flu going around your area next winter. Your doctor
says that you have a 10% chance (10 out of 100) of dying from this flu.*
*However, a new flu vaccine has been developed and tested. If taken, the vaccine prevents you from
catching the deadly flu.*
*However, there is one serious risk involved with taking this vaccine. The vaccine is made from a
somewhat weaker type of flu virus, and there is a 5% (5 out of 100) risk of the vaccine causing you to
die from the weaker type of flu.*
*Imagine that this vaccine is completely covered by health insurance. If you had to decide now, which
would you choose?*
Response scale: I would definitely not take the vaccine. I would thus accept the 10% chance

of dying from this flu. (1); I would probably not take the vaccine. I would thus accept the 10% chance of dying from this flu. (2); I would probably take the vaccine. I would thus accept the 5% chance of dying from the weaker flu in the vaccine (3); I would definitely take the vaccine. I would thus accept the 5% chance of dying from the weaker flu in the vaccine. (4)

Results:
Wave 2: Consistent with Stanovich and West (2008), we find that a significant proportion of participants ($Prop_{avoid} = 0.45$, 95% CI=[0.43,0.47]) displayed omission bias, i.e., chose to avoid the treatment (answer 1 or 2 vs. 3 or 4).
Wave 4: similar results ($Prop_{avoid} = 0.45$, 95% CI=[0.43,0.47]).

### A3.6.4   Probability matching vs. maximizing (Stanovich and West 2008)

Conditions: card problem, dice problem

Card problem: *Consider the following hypothetical situation:*
textitA deck with 10 cards is randomly shuffled 10 separate times. The 10 cards are composed of 7 cards with the number "1" on the down side and 3 cards with the number "2" on the down side.
*Each time the 10 cards are reshuffled, your task is to predict the number on the down side of the top card.*
*Imagine that you will receive \$100 for each downside number you correctly predict, and that you want to earn as much money as possible.*
*What would you predict after ...*
10 items: *shuffle #1,...shuffle #10*
Response scale (choose one): 1, 2
Measure: participant classified as using the MAX strategy (normative) if chose 1 ten times, the MATCH strategy if chose 1 seven times and 2 three times, and the OTHER strategy if made any other set of choices.

Dice problem: *Consider the following hypothetical situation:*
*Consider the following situation:*
*A die with 4 red faces and 2 green faces will be rolled 6 times.*
*Before each roll you will be asked to predict which color (red or green) will show up once the die is rolled.*
*Which color is most likely to show up after ...*
6 items: *roll #1,...roll #6*
Response scale (choose one): red, green
Measure: participant classified as using the MAX strategy (normative) if chose red six times, the MATCH strategy if chose red four times and green two times, and the OTHER strategy if made any other set of choices.

Results:
Wave 1: consistent with Stanovich and West (2008), we find that in both conditions, a significant proportion of participants chose a non-normative strategy ($P_{MAX}^{card} = 0.36$, 95% CI=[0.33,0.39]; $P_{MAX}^{dice} = 0.30$, 95% CI=[0.27,0.33]).
Wave 4: similar results ($P_{MAX}^{card} = 0.36$, 95% CI=[0.34,0.39]; $P_{MAX}^{dice} = 0.29$, 95% CI=[0.27,0.32]).

### A3.6.5   Dominator neglect (Stanovich and West 2008)

Question: *Assume that you are presented with two trays of black and white marbles, a large tray that contains 100 marbles and a small tray that contains 10 marbles. The marbles are spread in a single layer in each tray.*
*You must draw out one marble (without peeking, of course) from either tray. If you draw a black marble you win \$2. Consider a condition in which the small tray contains 1 black marble and 9*

*white marbles, and the large tray contains 8 black marbles and 92 white marbles. From which tray*
*would you prefer to select a marble in a real situation?*

Choice options: the small tray, the large tray

Results:

Wave 1: consistent with Stanovich and West (2008), we find that a significant minority of participants chose the non-normative large tray ($Prop_{large\ tray} = 0.36$, 95% CI=[0.34,0.38]).

Wave 4: similar results ($Prop_{large\ tray} = 0.38$, 95% CI=[0.36,0.40]).

## A3.7 Pricing study (Gui and Toubia 2023)

We replicate the study in Gui and Toubia (2023). See original paper for details. Table A4 lists the set of 40 products in the study. For each product, we vary the price from 0 to 200% of the regular price, in 20% increments. Each respondent answered one purchase intention question per product, with prices randomly drawn for each product and the order of products randomized across respondent. The wording of the question was as follows for *product* in *category* at *price*:

*Please consider the following product category: {category}.*
*Suppose you are in a grocery store, and you see the following product in that category: {product}.*
*The product is priced at: {price}.*
*Would you or would you not purchase the product?* (yes, I would purchase the product; No, I would not purchase the product).

Table A4: Categories, products and regular prices (from Gui and Toubia (2023))

| Category | Product | Price ($) |
|---|---|---|
| Fruit Juice | Capri Sun Variety Pack with Fruit Punch, Strawberry Kiwi & Pacific Cooler Juice Box Pouches, 30 ct Box, 6 fl oz Pouches | 9.43 |
| Fruit Drinks | Kool Aid Jammers Variety Pack with Tropical Punch, Grape & Cherry Kids Drink 0% Juice Box Pouches, 30 Ct Box, 6 fl oz Pouches | 7.27 |
| Baby Milk and Milk Flavoring | Horizon Organic Shelf-Stable Whole Milk Boxes, 8 oz., 12 Pack | 13.98 |
| Soup | Maruchan Ramen Noodle Chicken Flavor Soup, 3 Oz, 12 Count Shelf Stable Package | 9.97 |
| Cat Food - Wet Type | Purina Fancy Feast Chicken Feast Classic Grain Free Wet Cat Food Pate - 3 oz. Can | 0.88 |
| Pet Supplies - Dog Food | Purina Dog Chow Complete, Dry Dog Food for Adult Dogs High Protein, Real Chicken, 44 lb Bag | 29.17 |
| Snacks - Potato Chips | Lay's Classic Potato Snack Chips, Party Size, 13 oz Bag | 5.44 |
| Snacks - Tortilla Chips | Doritos Nacho Cheese Tortilla Snack Chips, Party Size, 14.5 oz Bag | 5.94 |
| Cereal - Ready to Eat | Cinnamon Toast Crunch Breakfast Cereal, Crispy Cinnamon Cereal, Family Size, 18.8 oz | 4.93 |
| Cookies | Little Debbie Oatmeal Creme Pies, 12 ct, 16.2 oz | 2.68 |
| Ground and Whole Bean Coffee | Folgers Classic Roast Ground Coffee, Medium Roast, 40.3-Ounce Canister | 13.24 |
| Soft Drinks - Carbonated | Coca-Cola Soda Pop, 12 fl oz, 12 Pack Cans | 8.26 |
| Bottled Water | OZARKA Brand 100% Natural Spring Water, 16.9-ounce plastic bottles (Pack of 35) | 19.96 |
| Candy - Chocolate | Hershey's Milk Chocolate Candy, Bars 1.55 oz, 6 Count | 6.48 |
| Candy - Non-Chocolate | HARIBO Goldbears Original Gummy Bears, 28.8oz Stand Up Bag | 6.48 |
| Soft Drinks - Low Calorie | Coca-Cola Zero Sugar Soda Pop, 16.9 fl oz, 6 Pack Cans | 5.18 |
| Frozen Italian Entrees | Smart Ones Three Cheese Ziti Marinara Frozen Meal, 9 Oz Box | 2.26 |
| Frozen Foods | Great Value All Natural Chicken Wing Sections, 4 lb (Frozen) | 12.98 |
| Ice Cream | Haagen Dazs Coffee Ice Cream, Gluten Free, Kosher, 14.0 oz | 4.18 |
| Frozen Novelties | Pop-Ice Assorted Fruit Freezer Ice Pops, Gluten-Free Snack, 1.5 oz, 80 Count Fruit Pops | 6.17 |
| Lunchmeat - Sliced - Refrigerated | Oscar Mayer Chopped Ham & Water product Deli Lunch Meat, 16 Oz Package | 4.33 |
| Frankfurters - Refrigerated | Oscar Mayer Classic Uncured Beef Franks Hot Dogs, 10 ct Pack | 3.94 |
| Refrigerated Bacon | Oscar Mayer Fully Cooked Original Bacon, 2.52 oz Box | 4.27 |
| Refrigerated Entrees | John Soules Foods Chicken Breast Fajita Strips, Refrigerated, 16oz, 18g Protein per 3oz Serving Size | 5.98 |
| Dairy Products | Land O Lakes Salted Stick Butter, 16 oz, 4 Sticks | 5.28 |
| Yogurt - Refrigerated | Chobani Non-Fat Greek Yogurt, Vanilla Blended 32 oz, Plastic | 5.58 |
| Refrigerated Deli Meats | Goya Cooked Ham 16 oz | 29.99 |
| Dairy - Milk - Refrigerated | Great Value Milk Whole Vitamin D Gallon | 3.92 |
| Bakery - Fresh Cakes | Little Debbie Zebra Cakes, 13 oz | 2.68 |
| Fresh Eggs | Eggland's Best Classic Extra Large White Eggs, 12 count | 3.18 |
| Fresh Fruit | Fresh Raspberries, 12 oz Container | 4.74 |
| Beer | Stella Artois Lager, 12 Pack, 11.2 fl oz Glass Bottles, 5% ABV, Domestic Beer | 15.73 |
| Light Beer (Low Calorie/Alcohol) | Bud Light Beer, 24 Pack, 12 fl oz Aluminum Cans, 4.2% ABV, Domestic Lager | 20.98 |
| Detergents - Heavy Duty - Liquid | Purex Liquid Laundry Detergent Plus OXI, Stain Defense Technology, 128 Fluid Ounces, 85 Wash Loads | 9.97 |
| Cleaning Supplies | ARM & HAMMER Pure Baking Soda, For Baking, Cleaning & Deodorizing, 1 lb Box | 1.54 |
| Toilet Tissue | Angel Soft Toilet Paper, 9 Mega Rolls, Soft and Strong Toilet Tissue | 6.68 |
| Paper Towels | Bounty Select-a-Size Paper Towels, 12 Double Rolls, White | 22.18 |
| Batteries | Duracell Coppertop AA Battery, Long Lasting Double A Batteries, 16 Pack | 15.97 |
| Pain Remedies - Headache | Tylenol Extra Strength Caplets with 500 mg Acetaminophen, 100 Ct | 10.97 |
| Cold Remedies -Adult | Equate Value Size Honey Lemon Cough Drops with Menthol, 160 Count | 4.68 |

