# OpenReview forum: "Twin-2K-500: A dataset for building digital twins of over 2,000 people based on their answers to over 500 questions"
_colmweb.org/COLM/2025/Workshop/Social_Sim — Social Sim'25_

### Official Review · Reviewer_uVx2 · 2025-07-20
**Review of Twin-2K-500**

**Rating:** 7
**Overall Assessment:** 4
**Confidence:** 4

**Review:**

This work presents a comprehensive dataset from recruiting and adminstering a wide range of measures to human participants. The collected data is an invaluable resource for researchers and this alone presents concrete, unique contributions to the community. The evaluation of different persona construction methods and their effectiveness in replicating the human responses (as digital twins) should ideally be expanded and dealt with greater rigor; however the value of this paper is already sufficiently presented by the newly collected dataset itself.

**Comments Suggestions And Typos:**

* The presentation of the digital twin evaluation study and result could have been improved. For example, showing a direct example of one of the methods or differentiating the three methods would have been helpful to understand what each of the methods represents.

* The tables and their captions could be made more detailed or easier to follow: for example, Table 2 could be restructured to better capture the results or some results could be omitted for a deeper dive of certain rows of interest. It was hard to follow what the checkmarks indicate.

**Ethical Concerns:**

The collected data encompasses survey item responses from human participants: the reviewer assumes authors followed standard best practices including guidelines subject to and in accordance with IRB guidelines.

In distributing the dataset, it will be imperative to discuss detailed plans to ensure data privacy and potential misuse. As this is a workshop (non-archival) submission, the reviewer does not explicitly require such details to be considered in the manuscript but expects authors to care for removal of any personally identifiable material from the data.

**Paper Summary:**

This work presents a dataset for evaluating (and training) large language models (LLMs) on emulating diverse aspects of human behavior, including items from psychology, economics, and cognitive science studies. A dataset consisting of comprehensive responses to 500 questions from over 2000 individuals is presented and made available for researchers.

**Relevance:**

4

**Summary Of Strengths:**

* The comprehensiveness of the Twin-2K-500 dataset, both in terms of the selection of administered survey items and the sheer number of recruited participants (cared to be representstive of the U.S. population) brings significance of this dataset as a basis for studying LLM emulation of human behavior

* The inclusion of an additional wave (Wave 4) for test-retest validation helps establish an important yet often understated aspect of human study data.

**Summary Of Weaknesses:**

* A more careful evaluation of digital twins or virtual LLM personas against the human data would have added greater value to the paper. The description and treatment of the experimental setup and results were hard to follow and seems insufficiently detailed, which is understandable given the page limit. The presented methods of constructing digital twins do not include more recent, advanced methods of using more detailed persona descriptions and backstories as presented in Park et al. 2024 [1] or Moon et al. 2024 [2].

* Following the point above, it would have been great to seen deepr analysis on whether the LLM digital twins show simillar-to-human levels of internal consistency across items that humans do show consistency. For example, items drawn from the same original study or battery, human participants typically show substantial levels of consistency as measured via Cronbach's alpha: do LLMs as well?

* The dataset presents individual human responses over a wide range of survey items: (1) in administering this long survey, what were the key considerations and issues that had to be addressed? Were there any new challenges aside from maintaining participant attention, such as question ordering effects? (2) what does the fact that we have such comprehensive data from an individual afford in research in constructing and evaluating LLM digital twins? This point is less a weakness but point that if addressed or developed by authors could greatly augment the presented work.

[1] Park, Joon Sung, et al. "Generative agent simulations of 1,000 people." arXiv preprint arXiv:2411.10109 (2024).

[2] Moon, Suhong, et al. "Virtual personas for language models via an anthology of backstories." arXiv preprint arXiv:2407.06576 (2024).

---

### Meta-Review · Area_Chair_XDgb · 2025-07-21

**Recommendation:** Accept

**Metareview:**

--